# Multiview Equivariance Improves 3D Correspondence Understanding with Minimal Feature Finetuning

**Yang You**[1], **Yixin Li**[1], **Congyue Deng**[1], **Yue Wang**[2], **Leonidas Guibas**[1,✉]

[1] Department of Computer Science, Stanford University, U.S.A.
[2] Department of Computer Science, University of Southern California, U.S.A.
✉ guibas@cs.stanford.edu
https://github.com/qq456cvb/3DCorrEnhance

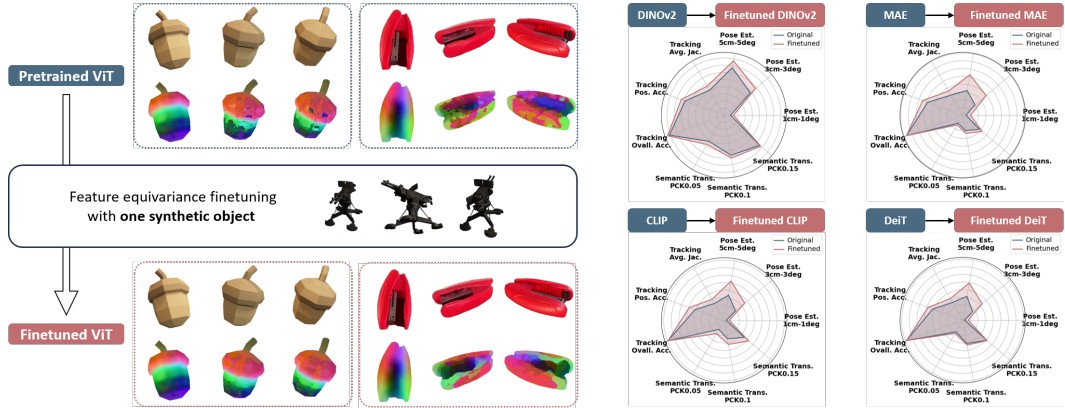

Figure 1: **Improving 3D correspondence understanding through finetuning on feature equivariance. Left:** finetuning feature equivariance on one synthetic object can already enhance the vision transformer's ability to generate better 3D feature correspondences on general objects. **Right:** This improvement further leads to superior performance across multiple 3D tasks, including pose estimation, video tracking, and semantic correspondence.

## Abstract

Vision foundation models, particularly the ViT family, have revolutionized image understanding by providing rich semantic features. However, despite their success in 2D comprehension, their abilities on grasping 3D spatial relationships are still unclear. In this work, we evaluate and enhance the 3D awareness of ViT-based models. We begin by systematically assessing their ability to learn 3D equivariant features, specifically examining the consistency of semantic embeddings across different viewpoints. Our findings indicate that improved 3D equivariance leads to better performance on various downstream tasks, including pose estimation, tracking, and semantic transfer. Building on this insight, we propose a simple yet effective finetuning strategy based on 3D correspondences, which significantly enhances the 3D correspondence understanding of existing vision models. Remarkably, finetuning on a single object for one iteration results in substantial gains.

## 1 Introduction

Common camera imaging systems struggle to depict the 3D world due to the limitation of capturing only a single perspective at any given moment. In contrast, human perceptual capabilities exhibit a remarkable trait known as view equivariance Köhler (1967); Koffka (2013); Wilson & Farah (2003), allowing us to robustly understand 3D spatial relationships, as seen in tasks ranging from basic object recognition Vetter et al. (1995); DiCarlo & Cox (2007) to more complex processes like mental rotation and simulation Stewart et al. (2022).

Current large vision models, however, are primarily trained on 2D images, owing to the ease of data acquisition and annotation in 2D. Consequently, their performance is typically evaluated on 2D tasks

Amir et al. (2021); Hedlin et al. (2023); Tang et al. (2023); Zhang et al. (2023). This raises critical questions: *To what extent do these models possess an inherent awareness of 3D structures? How does this awareness impact their performance on image-based 3D vision tasks? And, can we further enhance the 3D awareness of these vision foundation models?*

Many image-based 3D scene understanding and content generation tasks depend heavily on large 2D vision models, underscoring the importance of investigating these questions. Existing works have begun to explore this area in task-specific contexts. For example, DietNeRF Jain et al. (2021) finds that CLIP Radford et al. (2021) demonstrates higher feature similarities between views from the same scene than from different scenes, which aids 3D reconstruction. LeRF Kerr et al. (2023) shows that regularizing CLIP with DINO Caron et al. (2021) features improves 3D feature distillation from multiple views. However, these studies are tied to specific tasks such as feature distillation. El Banani et al. (2024) probes the multi-view consistency of ViTs on the NAVI Jampani et al. (2023) and ScanNet Dai et al. (2017) datasets. However, the limited size of these datasets makes it challenging to draw comprehensive conclusions.

To address the first question, *how well do vision models understand 3D structures*, we present a comprehensive study of the 3D awareness of large 2D vision models. Specifically, we investigate the *view equivariance* of latent features—i.e., the consistency of multi-view 2D image features representing the same 3D point across different views. Using off-the-shelf multiview correspondences rendered from Objaverse Deitke et al. (2023) (synthetic) and MVImgNet Yu et al. (2023) (real-world), we find that current large vision models do exhibit some degree of view-consistent feature generation, with DINOv2 demonstrating the strongest performance.

To answer the second question, *how does this awareness influence performance in image-based 3D vision tasks*, we find that the quality of 3D equivariance is strongly correlated with performance on three downstream tasks requiring 3D correspondence understanding: pose estimation, video tracking, and semantic correspondence. Consistent with previous findings Örnek et al. (2023); Tumanyan et al. (2024); Zhang et al. (2023), DINOv2 Oquab et al. (2023) excels in these tasks.

Finally, to address the third question, *can we improve the 3D awareness of vision foundation models*, we propose a simple yet effective method to enhance the view equivariance of 2D foundation models, thereby significantly improving their 3D correspondence understanding. During training, we randomly select two different views of the same object from Objaverse and sample corresponding pixels. We apply the SmoothAP Brown et al. (2020) loss to enforce feature similarity between these corresponding pixels. This finetuning process, requiring only 10K iterations with LoRA and an additional convolutional layer of a Vision Transformer (ViT), significantly improves the performance of all tested models on 3D tasks. For instance, DINOv2 gains improvements of **9.58** (3cm-3deg in pose estimation), **5.0** (Average Jaccard in tracking), and **5.06** (PCK@0.05 in semantic correspondence). Surprisingly, even finetuning on a single multi-view pair sampled from one object for just one iteration yields notable gains in 3D correspondence understanding. In such cases, DINOv2's performance improves by **4.85**, **3.55**, and **3.47** for 3cm-3deg (pose estimation), Average Jaccard (tracking), and PCK@0.05 (semantic correspondence), respectively.

To summarize, our key contributions are: (i) We conduct a comprehensive evaluation of 3D equivariance capabilities in 2D vision foundation models. (ii) We demonstrate that the quality of 3D equivariance is closely tied to performance on three downstream tasks that require 3D correspondence understanding: pose estimation, video tracking, and semantic correspondence. (iii) We propose a simple but effective finetuning method that improves the 3D correspondence understanding of 2D foundation models, leading to marked performance gains across all evaluated tasks.

## 2 EVALUATION OF MULTIVIEW FEATURE EQUIVARIANCE

To assess how effectively current vision transformers capture 3D correspondence understanding, we introduce a 3D equivariance evaluation benchmark focused on the quality of correspondences between 2D points across different views for the same object. Additionally, we present three well-established application tasks that rely on 3D correspondence, demonstrating a strong correlation between the quality of 3D equivariance and downstream task performance. We evaluate five state-of-the-art vision transformers: DINOv2 Oquab et al. (2023), DINOv2-Reg Darcet et al. (2023), MAE He et al. (2022a), CLIP Radford et al. (2021) and DeiT Touvron et al. (2022), extracting their final-layer features with L2 normalization. For DINOv2, we use the *base* model; results for other variants are provided in the supplementary material.

To evaluate 3D equivariance, we utilize rendered or annotated multiview correspondences from **Objaverse** Deitke et al. (2023) and **MVImgNet** Yu et al. (2023), covering both synthetic and real images. For Objaverse, we randomly select 1,000 objects from the Objaverse repository, rendered across 42 uniformly distributed camera views, producing 42,000 images. Dense correspondences are computed for each object across every unique ordered pair of views, resulting in 1.8 billion correspondence pairs for evaluation. Similarly, 1,000 objects are randomly drawn from MVImgNet, yielding 33.3 million annotated correspondence pairs for evaluation. Since MVImgNet employs COLMAP to reconstruct 3D points, it provides sparser correspondences compared to Objaverse.

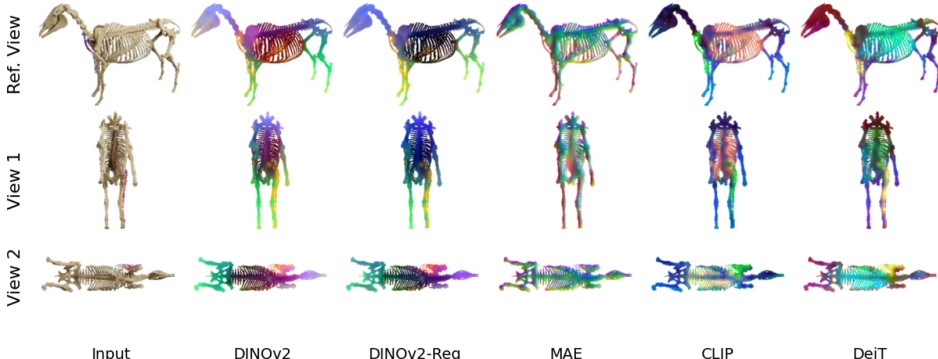

Figure 2: **Feature visualizations of different models.** The sample image is rendered from Objeverse. Colors are computed from the high-dimensional features using PCA. We can see that MAE struggles to distinguish different parts of the content (*e.g.*similar features between head and body). Both CLIP and DeiT produce inconsistent features for the chest region between View 1 and View 2. DINOv2 gives the best correspondence.

**Metric and Results** We propose the **Average Pixel Error%** (APE), a metric that quantifies the average distance between predicted and ground-truth pixel correspondences, normalized by the length of the shortest image edge. The predicted correspondence is determined by identifying the nearest neighbor in the second view, given a reference point feature in the first view. APE for Objaverse is shown in Figure 3, where APE is plotted on the x-axis, meaning lower values (towards the left) indicate better performance. APE and PCDP for MVImgNet are plotted on Figure 5's y-axis with hollow circle ○ and striped bar ⊠ representing the evaluted pretrained models (fine-tuning results will be discussed later). **Percentage of Correct Dense Points%** (PCDP) is a metric designed to evaluate dense correspondences, similar to Percentage of Correct Keypoints% (PCK). It is reported at various thresholds (5%, 10%, and 20% of the shortest image edge). We can see that DINOv2 and its registered version outperform other vision transformers, highlighting DINOv2's superior capability for 3D equivariance. In Figure 2, we provide feature visualizations using PCA, where DINOv2 again demonstrates the best multiview feature consistency.

## 2.1 FEATURE EQUIVARIANCE CORRELATES TO CERTAIN TASK PERFORMANCES

3D Equivariance itself is not interesting unless it can be used. Below, we will talk about three mature downstream applications that require 3D equivariance capability, and show a correlation between the quality of 3D equivariance and the downstream applications.

### 2.1.1 TASK DEFINITIONS

**One-Shot Object Pose Estimation** In one-shot pose estimation, we assume access to a video sequence or 3D mesh of the target object and aim to estimate its pose in arbitrary environments. During onboarding, we store dense 2D image features from all rendered or annotated views in a database. At inference, we compute correspondences between the input image and the stored features to match 2D keypoints in the image to their 3D counterparts. Pose estimation from these 2D-3D correspondences is achieved using RANSAC Fischler & Bolles (1981) PnP (Perspective-n-Point). Points are uniformly sampled using stratified sampling (stride 4) on $512 \times 512$ resized images. RANSAC PnP runs for 10,000 iterations with a threshold of 8.

We evaluate on the OnePose-LowTexture and YCB-Video datasets. OnePose-LowTexture He et al. (2022b) includes 40 low-textured household items captured in two videos: one for reference and

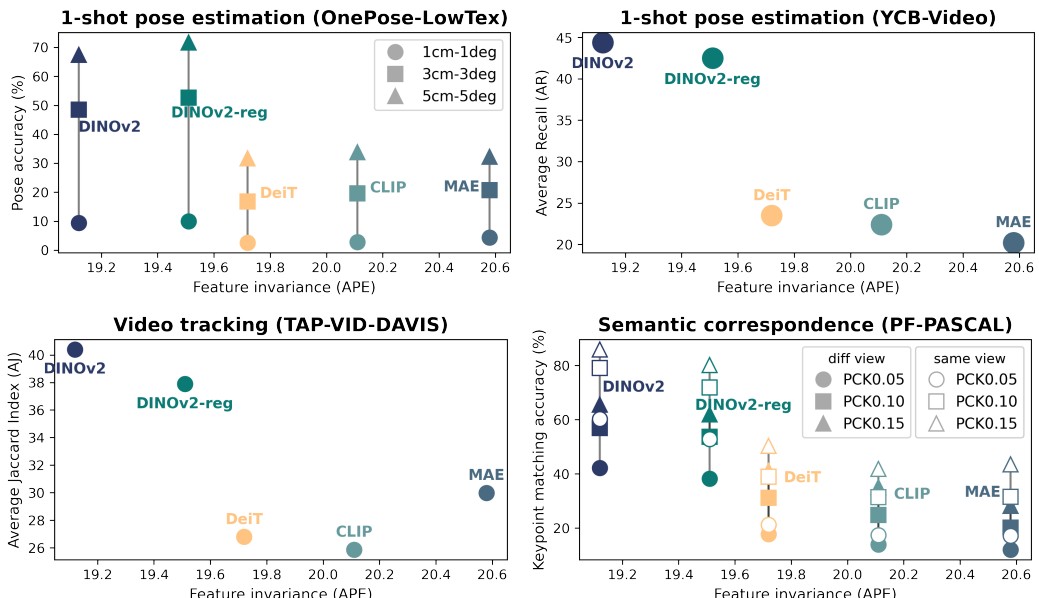

Figure 3: **Correlation between multiview feature equivariance and the task performances.** Along the horizontal axis, lower APE indicates better feature equivariance, while the vertical axis reflects higher task performance across all four plots. The data points align roughly along the diagonal from the top left to the bottom right, suggesting a strong correlation between improved feature equivariance and better task performance.

one for testing, simulating a one-shot scenario. We evaluate on every 10 frames in the video. Following He et al. (2022b), pose accuracy is evaluated using 1cm-1deg, 3cm-3deg, and 5cm-5deg thresholds. The YCB-Video dataset Xiang et al. (2017) comprises 21 objects and 92 RGB-D video sequences with pose annotations and CAD models for one-shot generalization. A database is created by rendering objects from 96 icospherical viewpoints. We report Average Recall (AR) for Visible Surface Discrepancy (VSD), Maximum Symmetry-Aware Surface Distance (MSSD), and Maximum Symmetry-Aware Projection Distance (MSPD) following Hodaň et al. (2020).

**Video Tracking** For video tracking, given the reference frame, we identify corresponding points in other frames by computing cosine similarities between the dense features of the target object. To improve robustness and accuracy, we follow the process in DINO-Tracker Tumanyan et al. (2024), which applies a softmax operation within the neighborhood of the location with highest similarity.

We evaluate the models on the TAP-Vid-DAVIS Doersch et al. (2022) dataset, a benchmark designed for testing video tracking in complex, real-world scenarios. Performance is measured using commonly applied metrics Tumanyan et al. (2024), including the Average Jaccard Index (AJ), Position Accuracy ($\delta_{avg}^x$), and Occlusion Accuracy (OA).

**Semantic Correspondence** In the semantic correspondence task, we utilize feature correspondences to establish precise keypoint matches between images captured from different instances from the same category. Following the method in Zhang et al. (2023), for a given reference keypoint, we identify the best match by selecting the location with the highest cosine feature similarity.

We use the PF-PASCAL Ham et al. (2017) dataset as our evaluation benchmark. This dataset typically consists of image pairs taken from the same viewpoint, but we additionally report the result by shuffling the image pairs to include different viewpoints, thereby increasing the challenge. We follow standard practice to use PCK@0.05, PCK@0.10, and PCK@0.15 as evaluation metrics.

The pipelines for all three tasks are illustrated in the figures provided in the supplementary material.

### 2.1.2 ON THE CHOICE OF THREE TASKS

Correspondence estimation is a fundamental component of 3D vision understanding, underlying key tasks such as epipolar geometry, stereo vision for 3D reconstruction, and optical flow or tracking to

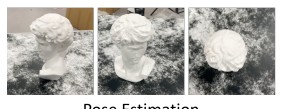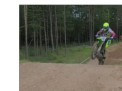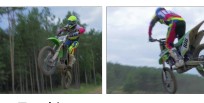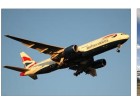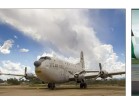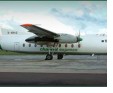

| Pose Estimation | Tracking | Semantic Transfer |
|---|---|---|

Figure 4: **Illustration of different types of correspondence tasks evaluated in our work.**

describe the motion of a perceived 3D world. Stereo cameras, and even human perception, rely on disparity maps—effectively, correspondences between projected 3D parts to understand depth and spatial relationships.

The three tasks we evaluated—pose estimation, video tracking, and semantic correspondence—are intentionally selected to cover diverse aspects of correspondence estimation, ranging from simpler to more complex scenarios: 1. **Pose Estimation** examines correspondences within the same instance under rigid transformations ($SE(3)$); 2. **Video Tracking** extends this to correspondences for the same instance under potential non-rigid or articulated transformations, such as humans or animals in motion; 3. **Semantic Correspondence** requires correspondences across different instances with similar semantics, often under arbitrary viewpoint changes. An qualitative illustration of these correspondence types is shown in Figure 4.

### 2.1.3 RESULTS AND FINDINGS

Quantitative results are presented in Figure 3, where the y-axis in each graph shows the performance of the vision models. DINOv2 consistently outperforms all other models across all three tasks, in alignment with the rankings for 3D equivariance on the x-axis. There is a clear correlation between the quality of 3D equivariance and performance on the downstream tasks: methods with lower APE tend to perform better across all tasks, clustering towards the top-left of the graphs.

## 3 FEATURE FINETUNING WITH MULTIVIEW EQUIVARIANCE

Given the correlation between the multiview equivariance of network features and task performances, we naturally come up with a question: *Can we finetune the networks on feature equivariance to improve their 3D correspondence understanding and achieve better task performances?*

**Finetuning method** The high-level intuition of improving the multiview equivariance of the network features is to enforce the similarity between features of corresponding pixels in 3D space. We experiment with multiple strategies including different training objectives and network architectures.

For the training loss, rather than employing a conventional contrastive loss, we opted for the SmoothAP Brown et al. (2020) loss, which demonstrated superior performance. While contrastive loss can help align the features of corresponding pixels, it relies on a predefined fixed margin for positive and negative samples, which is ad hoc and often suboptimal. In contrast, SmoothAP optimizes a ranking loss directly, leading to an improved average precision for feature retrieval between corresponding pixels. We also experimented with the differentiable Procrustes alignment loss Li et al. (2022), but it did not outperform. Detailed ablation results are given in Section 4.3.

In terms of architecture, we apply LoRA in the last four blocks to finetune large foundation models, we introduced a single convolutional layer with a kernel size of 3 and a stride of 1. The motivation behind this addition is rooted in the observation that ViT-family models process image tokens as patches, resulting in much lower-resolution feature maps (e.g., 14x smaller in DINOv2). The standard approach to obtain high-resolution per-pixel features is to apply linear interpolation. Consequently, it is beneficial to explicitly exchange information between neighboring patches before interpolation to achieve more accurate results. More ablation results are given in Section 4.1.

During training, we randomly select two views of the same object from a 10K subset of Objaverse at each iteration and sample corresponding pixels. The model is trained for 10K iterations using the AdamW optimizer with a learning rate of 1e-5 and weight decay of 1e-4. In the supplementary, we show that our finetuning method is robust to the choice of learning rate.

### 3.1 IMPROVED FEATURE EQUIVARIANCE WITH GENERALIZATION

Figure 5 illustrates the performance of various models before and after finetuning. After finetuning on Objaverse, all models show improved 3D equivariance on both Objaverse (synthetic) and

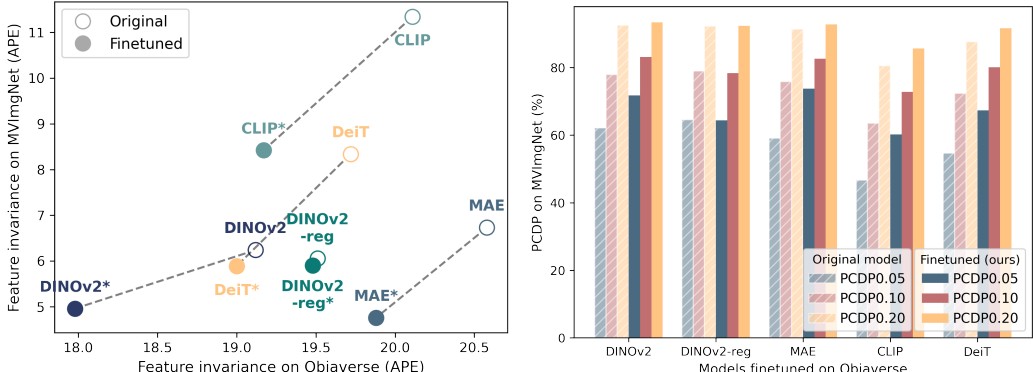

Figure 5: **Generalization from synthetic images (Objaverse) to real images (MVImgNet). Left:** Data points roughly around the diagonal from the bottom left to the upper right indicate the correlation between the APE tested on the two datasets. The * next to the model name means it is finetuned. All finetuning is done on Objaverse with only synthetic data. **Right:** Finetuned on Objaverse, the feature equivariance of the model (measured in PCDP) improves on MVImgNet.

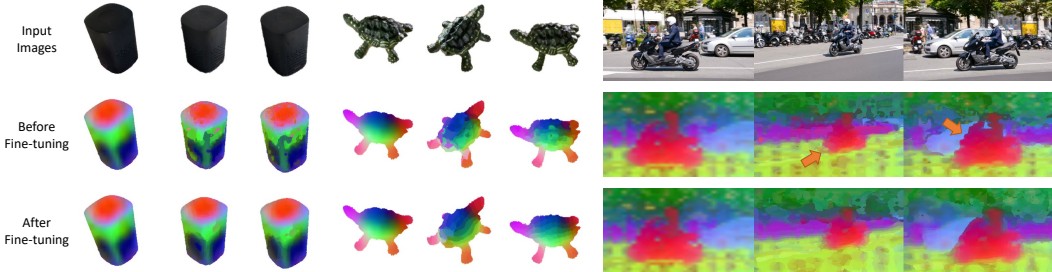

Figure 6: **Feature visualization of DINOv2 before and after finetuning on MVImgNet objects (left two) and TAP-VID-DAVIS scenes (right one).** For each example, we select three different views. The first column provides a reference color produced by PCA, while the second and third columns show the predicted feature correspondences. Our finetuned model demonstrates reduced noise and smoother feature boundaries, particularly noticeable in the reduction of jagged edges.

MVImgNet (real-world). This demonstrates the capacity of vision foundation models to perform sim-to-real transfer, as finetuning on synthetic Objaverse objects results in enhanced performance on the real-world MVImgNet dataset. Additionally, the performance on the two datasets is correlated, with data points roughly aligning along the diagonal, indicating that improvements in synthetic environments translate well to real-world settings. DINOv2 stands out as the best model. We also compare the feature visualizations before and after finetuning in Figure 6, from which we can see that after finetuning the model produces more consistent features with less noise.

## 3.2 IMPROVED TASK PERFORMANCES

**One-shot Object Pose Estimation** Figure 7 shows the performance of pose estimation on the OnePose-LowTex and YCB-Video datasets before and after fine-tuning. As illustrated, all Vision Transformers (ViTs) exhibit noticeable improvements after being fine-tuned on synthetic Objaverse data. For instance, the best-performing model, DINOv2-Reg, improves by **3.46**, **6.67**, and **6.92** for the 1cm-1deg, 3cm-3deg, and 5cm-5deg thresholds, respectively. Additionally, models that performed weaker before fine-tuning show larger gains. For example, DeiT improves by **4.65**, **16.39**, and **17.76**. Similar trends are observed for the YCB-Video dataset, where models like MAE, initially the weakest, show substantial improvement after fine-tuning.

**Video Tracking** Similarly, in the video tracking task, we observe consistent improvements across all ViTs after fine-tuning, as shown in Figure 8. The top-performing model, DINOv2, achieves improvements of **6.45**, **5.73**, and **2.69** in AJ, $\delta_{avg}^x$, and OA, respectively.

**Semantic Correspondence** In the semantic correspondence task, shown in Figure 9, DINOv2 exhibits improvements of **5.06**, **3.86**, and **1.98** for PCK@0.05, PCK@0.10, and PCK@0.15, respec-

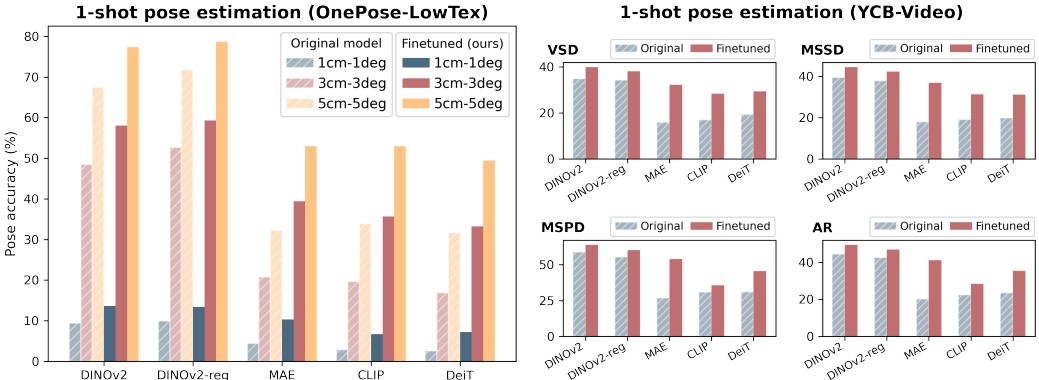

Figure 7: **One-shot pose estimation results before and after feature equivariance finetuning.**

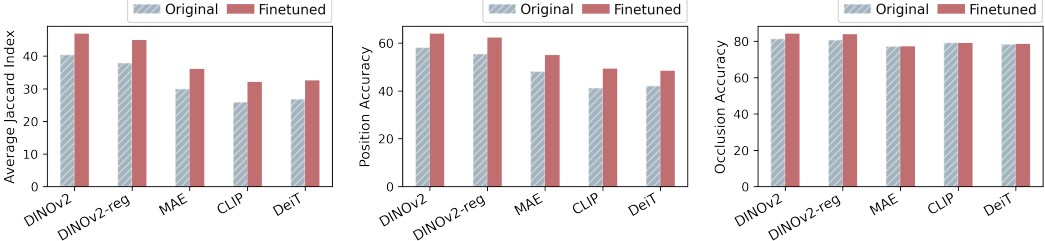

Figure 8: **Video tracking results before and after feature equivariance finetuning.**

tively. Notably, we find that fine-tuned models show enhanced understanding of keypoint semantics across different instances, even from the same viewpoint. This suggests that 3D equivariance contributes to a better understanding of fine-grained semantics, despite not finetuned for that purpose.

We also compared with FiT Yue et al. (2024) and DUSt3R Wang et al. (2024), while their performance are much worse than ours. Detailed quantitative results including FiT and DUSt3R on all these tasks are available in the supplementary materials.

### 3.3 EXTREMELY FEW-SHOT FINETUNING

**Training with Only One Object** We plot the performance relative to the number of training objects used, as shown in Figure 10, keeping the total number of iterations fixed at 10K. Surprisingly, fine-tuning on just one object already provides significant performance improvements. Additionally, the object was randomly selected from Objaverse. We tested six different objects, all of which yielded similar results. The results are shown in Figure 11. Notably, even simple shapes like an untextured hemisphere can enhance the 3D correspondence understanding of the ViTs in these tasks.

**Convergence Within a Few Iterations** Figure 12 plots the performance of downstream tasks versus the number of training iterations on a single object. Interestingly, our experiments reveal that training with just a single multi-view pair of one object for a single iteration significantly boosts the model's 3D equivariance, as shown by the sharp improvement at the first elbow of Figure 12. This finding is remarkable, indicating that fine-tuning for 3D correspondence in vision transformers is highly efficient in capturing essential 3D spatial relationships with minimal data. Even with such a minimal training setup, the model effectively learns the desired 3D properties, substantially improving performance across tasks without requiring extensive training or large datasets.

### 3.4 FINETUNING ViT ENHANCES 3D TASKS IN THE WILD

A key advantage of the ViT features studied here are highly generalizable across diverse datasets and tasks, supporting a even wider range of applications. For example, SparseDFF Wang et al. (2023) uses DINO to aggregate and fine-tune consistent features across views for few-shot transfer manipulation policy learning; LERF Kerr et al. (2023) employs dense DINO features for regularization; and Wild Gaussians Kulhanek et al. (2024) utilizes off-the-shelf DINO features as priors to estimate occlusions and reconstruct 3D scenes in complex settings. To demonstrate the effectiveness of our fine-tuned features, we conducted experiments on Wild Gaussians (W-G) and found that replacing the original features with our fine-tuned DINO features improved novel view synthesis quality in

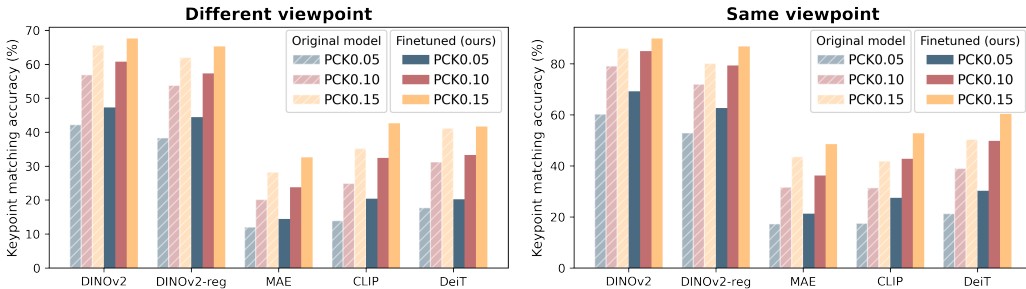

Figure 9: **Semantic correspondence results before and after feature equivariance finetuneing.**

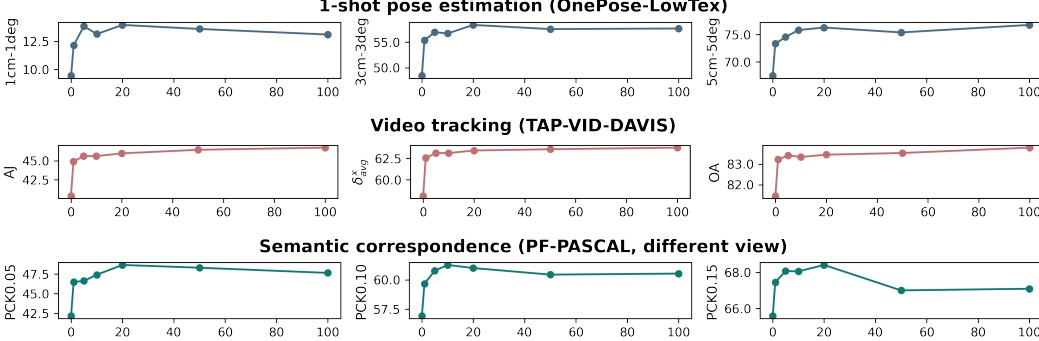

Figure 10: **Finetuned performances *w.r.t.* #training objects.** We evaluate the performances of the DINOv2 model finetuned with 0, 1, 5, 10, 20, 50, 100 objects on the three tasks.

the wild, as shown in Table 1. Additionally, in the supplementary we show that substituting LERF's DINO regularizer with our fine-tuned version enhances language-embedded field performance, with detailed results and analysis provided therein.

| | Mountain | | | Fountain | | | Corner | | | Patio | | | Spot | | | Patio-High | | |
|---|---|---|---|---|---|---|---|---|---|---|---|---|---|---|---|---|---|---|
| | PSNR↑ | SSIM↑ | LPIPS↓ | PSNR↑ | SSIM↑ | LPIPS↓ | PSNR↑ | SSIM↑ | LPIPS↓ | PSNR↑ | SSIM↑ | LPIPS↓ | PSNR↑ | SSIM↑ | LPIPS↓ | PSNR↑ | SSIM↑ | LPIPS↓ |
| W-G | 20.82 | 0.668 | 0.239 | 20.90 | 0.668 | 0.213 | 23.51 | 0.810 | 0.152 | **21.31** | 0.802 | 0.134 | 23.96 | 0.777 | 0.165 | 22.04 | 0.734 | 0.202 |
| Ours | **21.01** | **0.672** | **0.234** | **20.97** | **0.672** | **0.212** | **23.74** | **0.810** | **0.151** | 21.23 | 0.802 | **0.133** | **24.01** | **0.778** | **0.163** | **22.11** | 0.734 | **0.201** |

Table 1: **Quantitative comparison of novel view synthesis quality across different scenes.** Our fine-tuned DINO features consistently improve performance over the original Wild-Gaussians method, showing higher PSNR and SSIM scores, and lower LPIPS values.

## 4 DESIGN CHOICES FOR FINETUNING

In this section, we ablate and verify the design choices of our finetuning strategy and share some findings. We use the best DINOv2 *base* model for all our ablations.

### 4.1 ADDITIONAL CONVOLUTION LAYER HEAD

We append a single convolution layer to the original model architecture and find that gives surprisingly good performance. Adding a single convolutional layer to the finetuning architecture was motivated by the need to improve the resolution and consistency of the dense feature maps produced by Vision Transformer (ViT) models. The typical ViT models process images as low-resolution patches, and while global attention mechanisms facilitate communication between patches, they are not optimized for generating dense per-pixel features during interpolation. By incorporating a convolutional layer with a kernel size of 3 and a stride of 1, we can explicitly exchange information between neighboring patches, allowing the model to generate more accurate and high-resolution feature maps before interpolation. We ablate the number of convolutional layers and table 2 shows that one conv layer gives the best performance.

### 4.2 TRAINING DATA

**MVImgNet v.s. Objaverse** Our results indicate that finetuning on MVImgNet is slightly worse compared to finetuning on Objaverse, likely due to the denser correspondences provided by Ob-

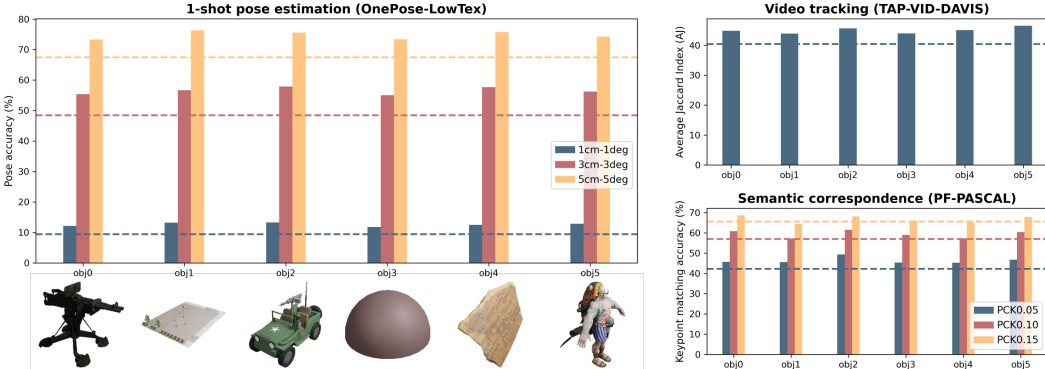

Figure 11: **Finetuning with different objects.** All results are tested with finetuned DINOv2. Dashed lines indicate the performances of the original pretrained model. The feature finetuning method is effective with as few as one single object. It also shows insensitivity to the specific choice of the object, even if the object has limited textures or is uncommon in daily life.

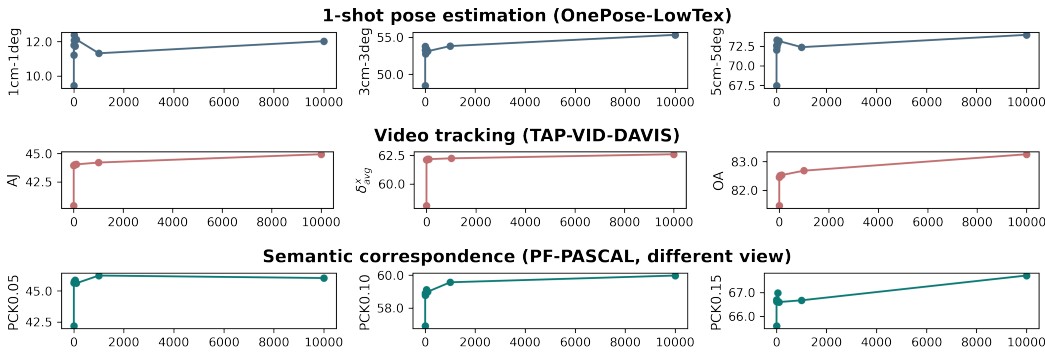

Figure 12: **Finetuned DINOv2 performances** *w.r.t.* **#training iterations, trained with** *only one object* **over 0, 1, 5, 10, 20, 50, 100, 1000, 10000 training iterations.**

javerse. Both datasets provide a similar object-centric multi-view setup. Although Objaverse is a synthetic dataset and MVImgNet consists of real-world captures, large foundation models tend to be largely agnostic to the distinction between simulated and real images.

**Object-centric datasets v.s. scene-centric datasets** An interesting result, as shown in Table 3, is that finetuning on scene-centric datasets (*e.g.* RealEstate10K Zhou et al. (2018), Spaces Flynn et al. (2019), and LLFF Mildenhall et al. (2019), which contain diverse real-world scenes with complex backgrounds, does not necessarily improve the performance but sometimes make it worse (*e.g.* PF-PASCAL). This may indicate that 3D objects themselves have already encoded enough 3D spatial reasoning information. And scene-centric dataset does include much more background clutter that may distract the network, leading to less accurate feature representations.

### 4.3 LOSS FUNCTIONS

We start with naive contrastive loss and found that it does not perform as well. This is because contrastive loss does not directly optimize for the correspondence. In contrast, SmoothAP optimizes a ranking loss directly, leading to an improved average precision for feature retrieval between corresponding pixels. We also experimented with the differentiable Procrustes alignment loss from Li et al. (2022), but it did not outperform SmoothAP. Detailed comparisons are given in Table 4.

## 5 RELATED WORKS

Vision Transformers Dosovitskiy (2020) (ViTs) have made significant strides in image understanding by employing self-attention mechanisms to capture global contextual information, outperforming traditional convolutional neural networks (CNNs) in tasks such as image classification and object detection. However, despite their success in 2D applications, adapting these models to grasp 3D spatial relationships remains a challenging and relatively unexplored area.

| ViT models | OnePose-LowTex | | | TAP-VID-DAVIS | | | PF-PASCAL (Diff. View) | | |
|---|---|---|---|---|---|---|---|---|---|
| | 1cm-1deg | 3cm-3deg | 5cm-5deg | AJ | $\delta^x_{avg}$ | OA | PCK0.05 | PCK0.10 | PCK0.15 |
| DINOv2-FT (Conv 0) | 11.69 | 53.85 | 72.83 | 44.50 | 60.79 | 84.08 | 44.82 | 57.14 | 65.26 |
| DINOv2-FT (Conv 1) | **13.58** | **58.03** | **77.35** | 46.85 | **63.84** | **84.15** | **47.25** | **60.76** | **67.57** |
| DINOv2-FT (Conv 2) | 13.12 | 56.14 | 75.45 | **47.42** | 63.25 | 84.12 | 46.32 | 58.05 | 64.90 |
| DINOv2-FT (Conv 3) | 12.15 | 53.63 | 74.46 | 46.84 | 62.14 | 82.90 | 41.60 | 53.97 | 60.22 |

Table 2: **Ablation on the number of appended conv layers.**

| ViT models | OnePose-LowTex | | | TAP-VID-DAVIS | | | PF-PASCAL (Diff. View) | | |
|---|---|---|---|---|---|---|---|---|---|
| | 1cm-1deg | 3cm-3deg | 5cm-5deg | AJ | $\delta^x_{avg}$ | OA | PCK0.05 | PCK0.10 | PCK0.15 |
| DINOv2-FT (Objaverse) | 13.58 | 58.03 | **77.35** | 46.85 | **63.84** | **84.15** | **47.25** | **60.76** | **67.57** |
| DINOv2-FT (MVImgNet) | 13.65 | 56.98 | 74.61 | 41.53 | 58.89 | 82.67 | 45.13 | 57.93 | 65.40 |
| DINOv2-FT (Scene-Centric) | **15.95** | **60.79** | 76.35 | **47.36** | 63.07 | 80.27 | 41.73 | 52.33 | 60.33 |

Table 3: **Ablation on the dataset used for finetuning.**

| ViT models | OnePose-LowTex | | | TAP-VID-DAVIS | | | PF-PASCAL (Diff. View) | | |
|---|---|---|---|---|---|---|---|---|---|
| | 1cm-1deg | 3cm-3deg | 5cm-5deg | AJ | $\delta^x_{avg}$ | OA | PCK0.05 | PCK0.10 | PCK0.15 |
| DINOv2-FT (SmoothAP) | **13.58** | **58.03** | **77.35** | **46.85** | **63.84** | **84.15** | **47.25** | **60.76** | **67.57** |
| DINOv2-FT (Contrastive) | 13.28 | 55.57 | 75.68 | 43.79 | 62.20 | 81.84 | 46.70 | 58.08 | 66.21 |
| DINOv2-FT (DiffProc) | 12.92 | 55.00 | 74.86 | 43.60 | 61.32 | 82.74 | 43.89 | 57.22 | 64.66 |

Table 4: **Ablation on the loss function used.** SmoothAP delivers the best overall performance.

There is growing interest in assessing the 3D comprehension of vision models. While some studies have investigated how well generative models capture geometric information from a single image Bhattad et al. (2024); Du et al. (2023); Sarkar et al. (2024), these efforts are generally specific to generative models, limiting their applicability to broader vision tasks. More closely aligned with our work is El Banani et al. (2024), which evaluated the 3D awareness of visual foundation models through task-specific probes and zero-shot inference using frozen features. In contrast, we delve deeper and introduce a simple yet effective method for finetuning 3D awareness in ViTs.

Several researchers have also explored applying large-scale models to 3D tasks. For instance, some approaches utilize features from pre-trained models for tasks such as correspondence matching Zhang et al. (2023); Cheng et al. (2024) and pose estimation Örnek et al. (2023). ImageNet3D Ma et al. (2024) investigates how global tokens from ViT vary across views to aid pose estimation. While their work focuses on view-dependent global features, ours emphasizes dense, pixel-level features invariant to viewpoint changes. Their top-down pose estimation approach classifies poses using pretrained features with a domain-specific linear layer, which limits its applicability across diverse datasets. In contrast, we argue that finding correspondences, or learning equivariant representations, is a more effective strategy for general unseen tasks and datasets.

Recent works, such as FiT Yue et al. (2024) and DVT Yang et al. (2024), attempt to finetune pre-trained features. FiT lifts 2D features into 3D space and then projects them back into 2D to enforce 3D consistency. DVT, on the other hand, implements a denoising process to reduce periodic noise artifacts in images, a method that is orthogonal to our approach. Additionally, DUSt3R Wang et al. (2024) directly predicts 3D coordinates for each 2D pixel, but it lacks a shared consistent feature space and forfeits the rich semantic information provided by large vision models.

# 6 CONCLUSION

In this work, we systematically evaluated the 3D awareness of large vision models, with a specific focus on their ability to maintain view equivariance. Our comprehensive study demonstrates that current vision transformers, particularly DINOv2, exhibit strong 3D equivariant properties, which significantly correlate with performance on downstream tasks such as pose estimation, video tracking, and semantic transfer. Building on these insights, we introduced a simple yet effective finetuning method that enhances the 3D correspondence understanding of 2D ViTs. By leveraging multiview correspondences and applying a loss function that enforces feature consistency across views, our approach yields substantial improvements in task performance with minimal computational overhead. Remarkably, even a single iteration of finetuning can lead to notable performance gains.

Our findings highlight the importance of 3D equivariance in vision models and provide a practical path to improving 3D correspondence understanding in existing models. We believe this work opens up new opportunities for enhancing the 3D capabilities of vision transformers. All code and resources will be made publicly available to support further research in this direction.

## 7 STATEMENTS

**Ethics Statement.** Our method leverages open-sourced simulation data and real data whose data collection process follows strict ethical guidelines. In using these data, we follow the same ethical considerations to protect sensitive information. There is no ethical concerns detected of the proposed method to our knowledge, and we will strive to adhere to ICLR code of conduct for future use of the proposed method.

**Reproducibility Statement.** We provide extensive details for ease of re-implementation. We strive to ensure our method is reproducible and the findings in this paper are generalizalble. We will release code, results, and scripts for reproduction to promote future research in 3D deep learning.

## 8 ACKNOWLEDGEMENTS

Yang You is supported in part by the Outstanding Doctoral Graduates Development Scholarship of Shanghai Jiao Tong University. Congyue Deng, Yang You, and Leonidas Guibas acknowledge support from the Toyota Research Institute University 2.0 Program, ARL grant W911NF-21-2-0104, a Vannevar Bush Faculty Fellowship, and a gift from the Flexiv corporation. Yue Wang acknowledges funding supports from Toyota Research Institute, Dolby, and Google DeepMind. Yue Wang is also supported by a Powell Research Award.

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
