# Multiview Equivariance Improves 3D Correspondence Understanding with Minimal Feature Finetuning – Supplementary Material

## A  Appendix

### A.1  Metric and Loss Implementation Details

In this section, we give the detailed mathematical definitions of the evaluation metrics and the loss used in our method.

- **Average Pixel Error (APE)**: Suppose we have $N$ objects, each rendered from $k = 42$ different views. For a pixel $x_1$ in the first image, the ground-truth corresponding pixel $x_2$ in the second image is determined via back-projection into 3D and re-rendering, excluding occluded points. The evaluated method predicts $\tilde{x}_2$. APE is computed as:

$$APE = \sum_N \sum_i^k \sum_j^k \sum_{x_1 \to x_2} \frac{\|x_2 - \tilde{x}_2\|_2}{\min(W, H)}$$

  where $W, H$ are the image width and height.

- **Percentage of Correct Dense Points (PCDP)**: PCDP measures the proportion of predicted points $\tilde{x}_2$ that fall within a normalized threshold $\delta$ of the ground-truth point $x_2$:

$$PCDP = \sum_N \sum_i^k \sum_j^k \sum_{x_1 \to x_2} \infty(\frac{\|x_2 - \tilde{x}_2\|_2}{\min(W, H)} < \delta)$$

  Here $\infty(\cdot)$ is the indicator function and $\delta$ is a threshold (commonly 0.05, 0.1 or 0.15).

- **Smooth Average Precision (SmoothAP)**: SmoothAP is used as the training loss to enforce accurate feature correspondences:

$$SmoothAP = \frac{1}{S_P} \sum_{i \in S_P} \frac{1 + \sum_{j \in S_P} \sigma(D_{ij})}{1 + \sum_{j \in S_P} \sigma(D_{ij}) + \sum_{j \in S_N} \sigma(D_{ij})}$$

  where given a query point $x_1$, $S_P$ is the positive set containing ground-truth points $\{x_2\}$, $S_N$ is the negative set containing all other points in the second view, and $\sigma$ is the sigmoid function, and $D_{ij} = f_j \cdot f_{x_1} - f_i \cdot f_{x_1}$ measures the difference in feature similarity with respect to the query point $x_1$. Ideally, we want all negative points to have smaller similarities with respect to $x_1$ than all positive ones. In this case, $\sum_{j \in S_N} \sigma(D_{ij}) = 0$ and we get $SmoothAP = 1$. In training, we optimize the loss: $1 - SmoothAP$.

### A.2  Quantitative Results on Objaverse and MVImgNet

The detailed quantitative results on 3D equivariance of Objaverse and MVImgNet are given in Table 1 and 2.

### A.3  Quantitative and Qualitative Results on the Three Tasks

We present detailed quantitative results for the three tasks (pose estimation, video tracking, and semantic transfer) in this section. Additionally, we compare our method with DUSt3R Wang et al. (2024), FiT Yue et al. (2024) and FiT-Reg Yue et al. (2024). FiT-Reg is FiT finetuned on DINOv2 with registers Darcet et al. (2023). For pose estimation and tracking, we also provide comparisons with state-of-the-art methods such as OnePose++ He et al. (2022b), MegaPose Labbé et al. (2022),

| Model | PCDP(%) | | | APE(%)↓ |
|---|---|---|---|---|
| | 0.05↑ | 0.1↑ | 0.2↑ | |
| DINOv2 Oquab et al. (2023) | 22.60 | 36.84 | 58.88 | 19.12 |
| finetuned | **30.61** | **43.65** | **61.78** | **17.98** |
| DINOv2-Reg Darcet et al. (2023) | **23.05** | **37.24** | **58.23** | 19.51 |
| finetuned | 22.81 | 36.39 | 57.84 | **19.48** |
| MAE He et al. (2022a) | 16.25 | 30.71 | 55.46 | 20.58 |
| finetuned | **22.57** | **35.94** | **56.93** | **19.88** |
| CLIP Radford et al. (2021) | 17.05 | 33.00 | 57.17 | 20.11 |
| finetuned | **22.54** | **38.01** | **59.71** | **19.17** |
| DeiT Touvron et al. (2022) | 18.07 | 33.89 | 58.05 | 19.72 |
| finetuned | **23.39** | **38.47** | **59.95** | **19.00** |

Table 1: **Evaluation of dense correspondence on Objaverse.**

| Model | PCDP(%) | | | APE(%)↓ |
|---|---|---|---|---|
| | 0.05↑ | 0.1↑ | 0.2↑ | |
| DINOv2 Oquab et al. (2023) | 62.09 | 77.94 | 92.49 | 6.24 |
| finetuned | **71.74** | **83.12** | **93.41** | **4.96** |
| DINOv2-Reg Darcet et al. (2023) | **64.54** | **78.99** | 92.25 | 6.06 |
| finetuned | 64.35 | 78.38 | **92.36** | **5.90** |
| MAE He et al. (2022a) | 59.10 | 75.82 | 91.42 | 6.73 |
| finetuned | **73.76** | **82.58** | **92.75** | **4.76** |
| CLIP Radford et al. (2021) | 46.63 | 63.49 | 80.53 | 11.34 |
| finetuned | **60.23** | **72.78** | **85.69** | **8.42** |
| DeiT Touvron et al. (2022) | 54.63 | 72.36 | 87.64 | 8.34 |
| finetuned | **67.31** | **80.12** | **91.63** | **5.89** |

Table 2: **Evaluation of dense correspondence on MVImgNet.**

and Co-Tracker Karaev et al. (2023), which are specifically trained on these tasks. The results are summarized in Tables 3, 4, 5, 6, and 7.

Our experiments reveal that although FiT aims for 3D consistency, it significantly disrupts the semantics of certain parts, as shown in Figure 1. While this semantic disruption may be acceptable for FiT's original tasks like semantic segmentation and depth estimation—where an additional linear head can correct these issues—it becomes problematic for our tasks that require 3D-consistent, dense, pixel-level features. We hypothesize that FiT's poor performance stems from its naive approach to learning 3D consistency through an explicit 3D Gaussian field. When outliers or noise are present, the simple mean square error causes feature representations to shift toward these outliers.

### A.4 Quantitative Results for Other Variants of DINOv2

In addition to evaluating the DINOv2 *base* model, we tested our finetuning method on other variants, including *small*, *large*, and *giant*. Our method consistently yields improvements across almost all metrics for these model variants. The full results are presented in Table 8.

### A.5 Results for Other Foundation Models with Different Architectures.

In addition to ViT, we apply our method to other architectures like ConvNeXt and find that we can consistently improve its performance on downstream tasks as well as shown in Table 9. However, we've also observed that ConvNeXt features are not as good as those of modern ViTs. Nonetheless, we do expect and observe improvements in non-ViT based methods like ConvNeXt. This finding

| Method | OnePose-LowTex | | |
| --- | --- | --- | --- |
| | 1cm-1deg | 3cm-3deg | 5cm-5deg |
| OnePose++ He et al. (2022b) | 16.8 | 57.7 | 72.1 |
| DUSt3R Wang et al. (2024) | 2.88 | 16.61 | 26.79 |
| FiT Yue et al. (2024) | 1.05 | 9.18 | 16.52 |
| FiT-Reg Yue et al. (2024) | 3.44 | 23.51 | 37.68 |
| DINOv2 Oquab et al. (2023) | 9.43 | 48.45 | 67.45 |
| Finetuned | **13.58** | **58.03** | **77.35** |
| DINOv2-Reg Darcet et al. (2023) | 9.95 | 52.65 | 71.72 |
| Finetuned | **13.41** | **59.32** | **78.64** |
| MAE He et al. (2022a) | 4.41 | 20.76 | 32.27 |
| Finetuned | **10.27** | **39.37** | **52.97** |
| CLIP Radford et al. (2021) | 2.85 | 19.65 | 33.84 |
| Finetuned | **6.72** | **35.63** | **52.94** |
| DeiT Touvron et al. (2022) | 2.55 | 16.85 | 31.67 |
| Finetuned | **7.20** | **33.24** | **49.43** |

Table 3: **Quantitative results of one-shot pose estimation on OnePose-LowTex.**

| Method | VSD | MSSD | MSPD | AR |
| --- | --- | --- | --- | --- |
| MegaPose Labbé et al. (2022) | 53.5 | 59.7 | 72.8 | 62.0 |
| DUSt3R Wang et al. (2024) | 11.6 | 11.5 | 15.8 | 13.0 |
| FiT Yue et al. (2024) | 4.4 | 3.2 | 3.4 | 3.7 |
| FiT-Reg Yue et al. (2024) | 10.2 | 9.4 | 11.3 | 10.3 |
| DINOv2 Oquab et al. (2023) | 34.9 | 39.4 | 58.8 | 44.4 |
| Finetuned | **39.9** | **44.4** | **63.9** | **49.4** |
| DINOv2-Reg Darcet et al. (2023) | 34.2 | 37.9 | 55.4 | 42.5 |
| Finetuned | **38.1** | **42.3** | **60.0** | **46.8** |
| MAE He et al. (2022a) | 15.9 | 17.9 | 26.8 | 20.2 |
| Finetuned | **32.2** | **36.8** | **54.0** | **41.0** |
| CLIP Radford et al. (2021) | 17.0 | 19.1 | 31.0 | 22.4 |
| Finetuned | **28.3** | **31.3** | **35.6** | **28.3** |
| DeiT Touvron et al. (2022) | 19.4 | 19.8 | 31.2 | 23.5 |
| Finetuned | **29.4** | **31.1** | **45.6** | **35.4** |

Table 4: **Quantitative results of one-shot pose estimation on YCB-Video.**

is particularly interesting as it teaches us a valuable lesson: with relatively simple 3D fine-tuning, we can achieve even better 3D features than those obtained through pretraining on a vast set of unstructured 2D images.

### A.6 RESULTS ON OTHER SEMANTIC-RELATED TASKS

Here, we report results on other semantic-related tasks less focused on 3D understanding. As shown in Table 10, our finetuning method performs on par or slightly worse compared to baseline models in these tasks. We recon that these tasks do not benefit as much from the dense 3D equivariant features our method emphasizes, but rather from coarse, object-level global features. For instance, in tasks where a plane or side of a box should share the same semantic mask and depth, pixel-level dense features are unnecessary to achieve satisfactory results. Future work can be explored to enhance object-level global feature representation.

| Method | TAP-VID-DAVIS | | |
|---|---|---|---|
| | AJ | $\delta^x_{avg}$ | OA |
| Co-Tracker Karaev et al. (2023) | 65.6 | 79.4 | 89.5 |
| DUSt3R Wang et al. (2024) | 13.06 | 22.64 | 77.27 |
| FiT Yue et al. (2024) | 20.45 | 33.46 | 77.27 |
| FiT-Reg | 23.28 | 37.30 | 77.27 |
| DINOv2 Oquab et al. (2023) | 40.40 | 58.11 | 81.46 |
| Finetuned | **46.85** | **63.84** | **84.15** |
| DINOv2-Reg Darcet et al. (2023) | 37.89 | 55.43 | 80.77 |
| Finetuned | **44.91** | **62.23** | **83.85** |
| MAE He et al. (2022a) | 29.99 | 48.16 | **77.27** |
| Finetuned | **36.04** | **54.97** | **77.27** |
| CLIP Radford et al. (2021) | 25.86 | 41.17 | **79.28** |
| Finetuned | **32.13** | **49.31** | 79.09 |
| DeiT Touvron et al. (2022) | 26.80 | 42.06 | 78.45 |
| Finetuned | **32.55** | **48.41** | **78.49** |

Table 5: **Quantitative results of tracking on TAP-VID-DAVIS.**

| Method | PF-PASCAL | | |
|---|---|---|---|
| | PCK0.05 | PCK0.10 | PCK0.15 |
| DUSt3R Wang et al. (2024) | 4.70 | 8.21 | 13.01 |
| FiT Yue et al. (2024) | 13.10 | 23.99 | 33.45 |
| FiT-Reg Yue et al. (2024) | 22.39 | 36.45 | 45.27 |
| DINOv2 Oquab et al. (2023) | 42.18 | 56.90 | 65.59 |
| Ours | **47.24** | **60.76** | **67.57** |
| DINOv2-Reg Darcet et al. (2023) | 38.29 | 53.74 | 61.94 |
| Finetuned | **44.44** | **57.27** | **65.27** |
| MAE He et al. (2022a) | 11.98 | 20.16 | 28.16 |
| Finetuned | **14.45** | **23.79** | **32.56** |
| CLIP Radford et al. (2021) | 13.87 | 24.85 | 35.13 |
| Finetuned | **20.39** | **32.36** | **42.58** |
| DeiT Touvron et al. (2022) | 17.73 | 31.17 | 41.17 |
| Finetuned | **20.24** | **33.29** | **41.62** |

Table 6: **Quantitative results of PF-PASCAL (Different Viewpoints).**

**Instance Recognition** The objective for this task is to identify and differentiate individual object instances within a scene, even when multiple objects belong to the same class (e.g., recognizing distinct cars in a street scene). This task was evaluated using the Paris-H(ard) Radenović et al. (2018) dataset, with performance measured by mean Average Precision (mAP), which captures the precision-recall trade-off. Two probe training configurations were explored: one utilizing only the class token (Cls) and another concatenating patch tokens with the class token (Cls+Patch). Our fine-tuned model demonstrated performance on par with the DINOv2 baseline, achieving mAP scores of 76.23 for Cls and 75.43 for Cls+Patch.

**Semantic Segmentation** This task involves assigning a semantic label to each pixel in an image, thereby grouping regions based on their object class, without distinguishing between individual instances of the same class. The VOC2012 Everingham et al. (2010) dataset was used to evaluate this

| Method | PF-PASCAL | | |
|---|---|---|---|
| | PCK0.05 | PCK0.10 | PCK0.15 |
| DUSt3R Wang et al. (2024) | 2.64 | 8.01 | 15.00 |
| FiT Yue et al. (2024) | 13.96 | 27.42 | 37.39 |
| FiT-Reg Yue et al. (2024) | 26.47 | 45.74 | 55.32 |
| DINOv2 Oquab et al. (2023) | 60.22 | 79.05 | 85.95 |
| Finetuned | **69.16** | **84.94** | **89.82** |
| DINOv2-Reg Darcet et al. (2023) | 52.86 | 71.93 | 80.11 |
| Finetuned | **62.63** | **79.24** | **86.69** |
| MAE He et al. (2022a) | 17.16 | 31.52 | 43.54 |
| Finetuned | **21.26** | **36.16** | **48.52** |
| CLIP Radford et al. (2021) | 17.44 | 31.38 | 41.81 |
| Finetuned | **27.40** | **42.72** | **52.67** |
| DeiT Touvron et al. (2022) | 21.21 | 38.96 | 50.36 |
| Finetuned | **30.18** | **49.69** | **60.34** |

Table 7: **Quantitative results of PF-PASCAL (Same Viewpoint).**

| ViT models | OnePose-LowTex | | | TAP-VID-DAVIS | | | PF-PASCAL (Diff. View) | | |
|---|---|---|---|---|---|---|---|---|---|
| | 1cm-1deg | 3cm-3deg | 5cm-5deg | AJ | $\delta_{avg}^x$ | OA | PCK0.05 | PCK0.10 | PCK0.15 |
| DINOv2-S | 8.14 | 45.77 | 65.79 | 37.56 | 55.04 | 80.54 | 39.02 | 53.26 | **61.49** |
| Finetuned | **12.85** | **56.17** | **74.27** | **45.17** | **61.35** | **83.14** | **41.02** | **53.78** | 60.95 |
| DINOv2-L | 10.83 | 51.68 | 70.01 | 42.56 | 59.88 | 83.29 | 44.22 | 57.92 | 65.85 |
| Finetuned | **13.86** | **58.79** | **77.46** | **49.10** | **65.00** | **85.42** | **51.66** | **62.96** | **70.48** |
| DINOv2-G | 13.58 | 58.73 | 76.27 | 44.79 | 61.01 | 85.27 | 44.57 | 57.63 | 65.76 |
| Finetuned | **14.58** | **60.08** | **78.48** | **50.77** | **66.00** | **85.82** | **50.89** | **61.98** | **68.44** |
| DINOv2-S-reg | 10.25 | 49.04 | 68.83 | 34.61 | 52.21 | 79.35 | 31.30 | 45.47 | 54.73 |
| Finetuned | **12.25** | **56.69** | **75.66** | **40.53** | **58.50** | **81.14** | **38.78** | **52.08** | **59.26** |
| DINOv2-L-reg | 10.89 | 51.17 | 69.99 | 39.47 | 56.69 | 82.26 | 41.26 | 56.24 | 63.38 |
| Finetuned | **14.00** | **58.58** | **77.12** | **46.43** | **63.20** | **84.43** | **48.03** | **60.17** | **67.13** |
| DINOv2-G-reg | 11.14 | 53.84 | 72.28 | 41.39 | 58.62 | 83.09 | 40.94 | 53.84 | 61.87 |
| Finetuned | **14.24** | **59.88** | **79.19** | **47.93** | **64.43** | **85.38** | **47.36** | **59.20** | **66.55** |

Table 8: **Other dino variant results on OnePose-LowTex, TAP-VID-DAVIS, and PF-PASCAL.**

| | OnePose-LowTex | | | TAP-VID-DAVIS | | | PF-PASCAL (Diff. View) | | |
|---|---|---|---|---|---|---|---|---|---|
| | 1cm-1deg | 3cm-3deg | 5cm-5deg | AJ | $\delta_{avg}$ | OA | PCK0.05 | PCK0.10 | PCK0.15 |
| ConvNext-small | 3.25 | 13.46 | 21.39 | 15.98 | 26.08 | **74.72** | 10.32 | 16.30 | 22.17 |
| small-finetuned | **5.28** | **19.98** | **28.23** | **16.70** | **26.56** | 74.54 | **11.61** | **19.38** | **25.56** |
| ConvNext-base | 5.10 | 22.22 | 34.81 | 17.57 | 28.21 | **72.47** | 13.62 | 21.03 | 27.81 |
| base-finetuned | **8.05** | **32.69** | **46.41** | **18.53** | **28.48** | 71.24 | **15.64** | **25.37** | **32.13** |
| ConvNext-large | 4.71 | 25.33 | 36.48 | 19.43 | 30.24 | 73.71 | 11.05 | 17.57 | 24.19 |
| large-finetuned | **7.21** | **30.68** | **44.47** | **19.45** | **30.68** | **74.33** | **14.56** | **24.04** | **31.57** |

Table 9: **ConvNext finetuning results on OnePose-LowTex, TAP-VID-DAVIS, and PF-PASCAL.**

task, with performance metrics including mean Intersection over Union (mIoU) and mean Accuracy (mAcc). These metrics assess the overlap between predicted segmentation and ground truth, as well as pixel-wise classification accuracy. Our fine-tuned model achieved an mIoU of 82.65 and mAcc of 90.21, performing slightly below but comparable to DINOv2.

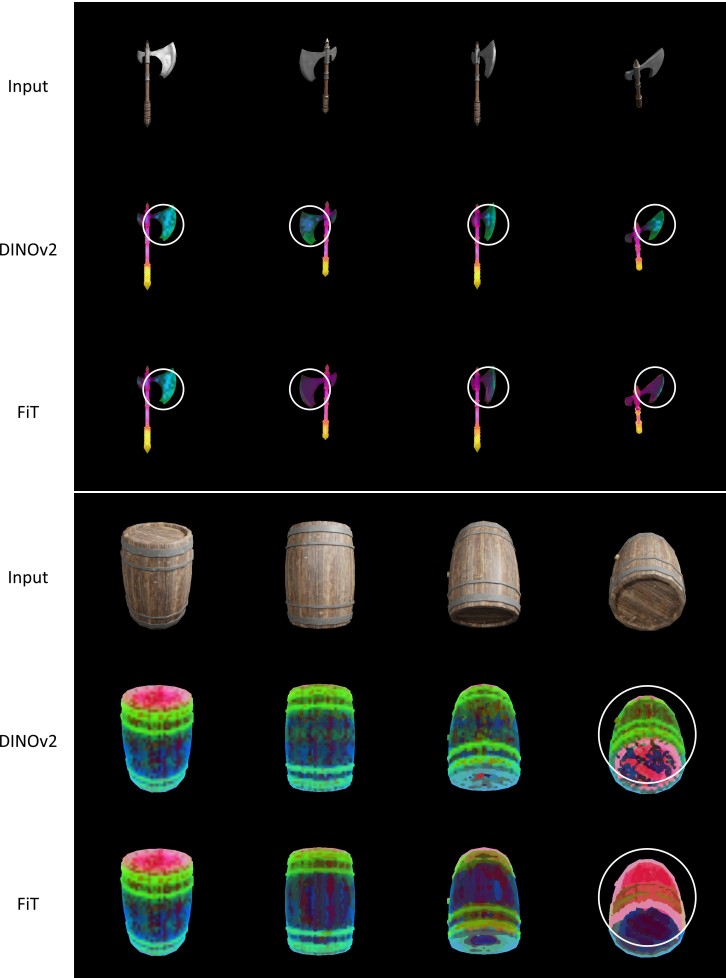

Figure 1: **FiT and DINOv2 semantic correspondence visualization. We find that FiT significantly disrupts the semantics of certain parts.**

**Depth Estimation** This task aims to predict the distance to each pixel in an image, effectively generating a depth map that represents the 3D structure of the scene. This task is critical for applications requiring spatial understanding, such as indoor navigation and scene reconstruction. We used the NYUv2 Silberman et al. (2012) dataset for evaluation, employing the $\delta_1$ accuracy and absolute relative error (abs rel) metrics to assess depth prediction performance. Our fine-tuned model achieved a $\delta_1$ score of 85.48 and an abs rel of 0.1299, slightly underperforming but comparable to the DINOv2 baseline.

| Model | Paris-H Inst. Recognition | | VOC2012 Segmentation | | NYUv2 Depth Estimation | |
|---|---|---|---|---|---|---|
| | Cls↑ | Cls+Patch↑ | mIoU↑ | mAcc↑ | $\delta_1$ ↑ | abs rel↓ |
| DINOv2 Oquab et al. (2023) | 75.92 | 73.69 | **83.60** | **90.82** | **86.88** | **0.1238** |
| Finetuned | **76.23** | **75.43** | 82.65 | 90.21 | 85.48 | 0.1299 |

Table 10: **Quantitative results of instance recoginition, semantic segmentation and depth estimation.**

## A.7 MORE RESULTS ON LERF

In addition to the Wild-Gaussians experiment in our main paper, we visualize LERF 3D features after replacing its DINO regularizer with our fine-tuned version in Figure 2. When given the text

| ViT models | OnePose-LowTex | | | TAP-VID-DAVIS | | | PF-PASCAL (Diff. View) | | |
|---|---|---|---|---|---|---|---|---|---|
| | 1cm-1deg | 3cm-3deg | 5cm-5deg | AJ | $\delta_{avg}^x$ | OA | PCK0.05 | PCK0.10 | PCK0.15 |
| DINOv2-FT (LR 1e-6) | 11.86 | 55.03 | 73.12 | 44.79 | 62.56 | 83.17 | **47.34** | 60.10 | **68.23** |
| DINOv2-FT (LR 3e-6) | 13.05 | 57.45 | 75.89 | 45.93 | 63.32 | 83.73 | 47.20 | 60.50 | 67.21 |
| DINOv2-FT (LR 1e-5) | **13.58** | 58.03 | 77.35 | **46.85** | **63.84** | **84.15** | 47.25 | **60.76** | 67.57 |
| DINOv2-FT (LR 3e-5) | 13.15 | **58.33** | **77.49** | 46.70 | 63.45 | 83.35 | 45.70 | 57.96 | 65.99 |

Table 11: **Ablation on the learning rate for finetuning.**

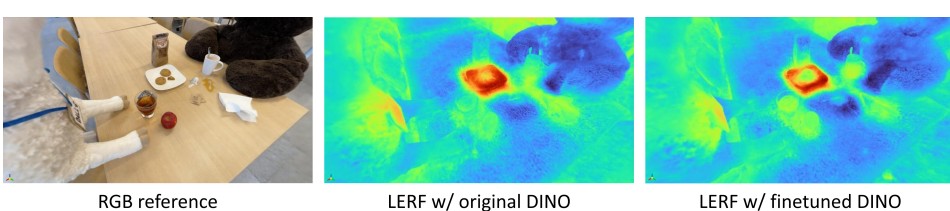

RGB reference         LERF w/ original DINO        LERF w/ finetuned DINO

Figure 2: **Visualization of LERF relevancy maps for the query "*plate*".** Our finetuned DINO features produce a more focused and accurate relevancy map compared to the original DINO features, with better localization of the plate region and reduced noise in irrelevant areas such as cookies.

query "plate", LERF with our fine-tuned DINO produced a better relevancy map than the original. Our relevancy map localizes of the plate region better and reduces noise in irrelevant areas such as cookies. These experiments demonstrate that our 3D fine-tuning produces better general-purpose features that enhance various applications.

## A.8 MORE ABLATION STUDY ANALYSIS

### A.8.1 ABLATIONS ON MULTI-LAYER FEATURE FUSION

In addition to extracting features solely from the last layer (11th), we experiment with two different variations: concatenating the features from the last 4 layers and concatenating features from the 2nd, 5th, 8th, and 11th layers. The results are presented in the Table 12. We find that fusing features from different layers does improve the instance-level correspondence a little bit but greatly harms semantic correspondences in tracking and semantic transfer. This indicates that features from earlier layers focus more on instance-level details, while the final layer captures more semantic information.

| | OnePose-LowTex | | | TAP-VID-DAVIS | | | PF-PASCAL (Diff. View) | | |
|---|---|---|---|---|---|---|---|---|---|
| | 1cm-1deg | 3cm-3deg | 5cm-5deg | AJ | $\delta_{avg}$ | OA | PCK0.05 | PCK0.10 | PCK0.15 |
| Layer 11 | 13.58 | 58.03 | 77.35 | **46.85** | **63.84** | **84.15** | **47.24** | **60.76** | **67.57** |
| Layer 2,5,8,11 | **15.34** | 59.56 | 76.81 | 39.67 | 56.74 | 76.29 | 39.84 | 53.05 | 60.15 |
| Layer 8,9,10,11 | 14.24 | **60.35** | **79.27** | 41.25 | 56.56 | 80.15 | 44.99 | 57.73 | 64.48 |

Table 12: **Ablations on the choice of feature blocks on DINOv2 base model.**

### A.8.2 ABLATIONS ON LEARNING RATE

Our finetuning method is insensitive to the choice of learning rate, and it can work within a reasonable range of learning rates, as shown in Table 11.

### A.8.3 QUALITATIVE RESULTS ON NUMBER OF CONVOLUTION LAYERS

Upon analyzing the effect of additional convolutional layers, we find that while one additional convolutional layer significantly improves the performance, adding two or three layers introduces noise into the features. This noise likely arises from the increased parameter freedom, which can overfit to local patterns and reduce the consistency of dense pixel-wise features, as shown in Figure 3. It clearly show that the additional layers produce less coherent features, leading to a degradation in downstream task performance.

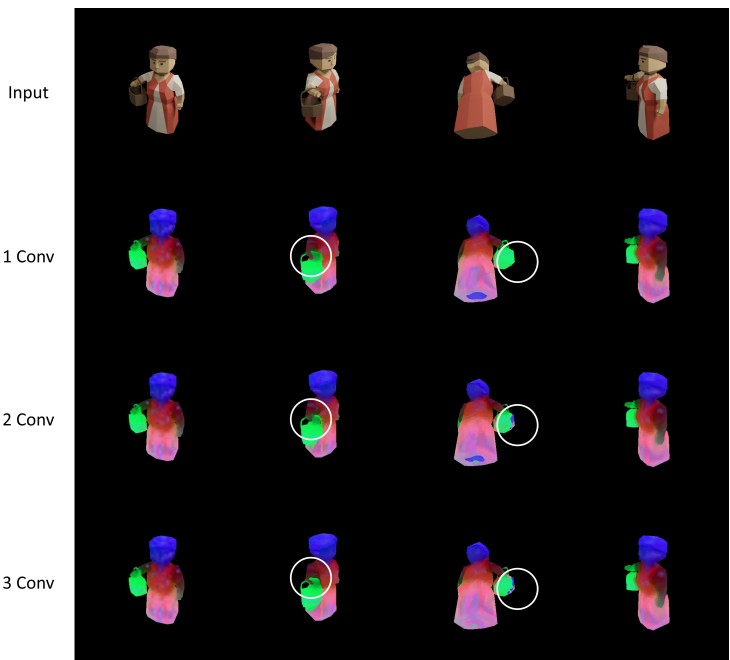

Figure 3: **Comparison of feature visualizations with varying convolutional layers.** Adding more than one convolutional layer introduces noise and reduces feature coherence, as shown by the highlighted regions.

## A.9 OTHER FINDINGS AND DISCUSSIONS

**Why Untextured Symmetric Hemisphere can Enhance 3D Understanding?** Unlike a perfect sphere, the hemisphere we used is not completely symmetric and provides information about edges and viewpoint orientation. Our visualization of the learned embeddings in Figure 4 shows that after fine-tuning on the hemisphere, the network achieves better edge correspondences and can differentiate between inward and outward views. Even though the object lacks texture, the shadows and edge features provide sufficient cues for the ViT features to develop 3D understanding.

Similarly, in cognitive science, scientists have discovered that the human brain excels at inferring 3D structure. Biederman's Recognition-by-Components (RBC) theory Biederman (1987) suggests that humans recognize objects through simple 3D primitives called geons (geometrical ions)—basic shapes such as cubes, cylinders, and cones.

**Training without Background Enhances Background-invariance** Interestingly, we observed that finetuning on object-centric datasets without backgrounds enhanced the foundation model's background invariance. Specifically, when comparing an object on a black background (i.e., no background) with the same object on a natural background from the same viewpoint, the finetuned model demonstrated superior feature consistency across corresponding pixels. We quantitatively validated this finding using pairs of images from a random 1K subset from the MSCOCO val dataset. For each annotated object, one image crop was masked while the other was unmasked. We measured the number of inliers by counting mutual nearest neighbors in the feature space that were within 1 pixel of the ground truth. The results confirmed that our finetuned model significantly improved feature consistency across these variations.

## A.10 PIPELINE VISUALIZATION FOR DOWNSTREAM APPLICATIONS

Figures 6, 7, 8, and 9 illustrate the detailed pipelines for various downstream tasks. Note that for pose estimation, tracking, and semantic transfer, no linear fine-tuning is applied. These tasks exclusively assess the quality of the pretrained features from the ViT.

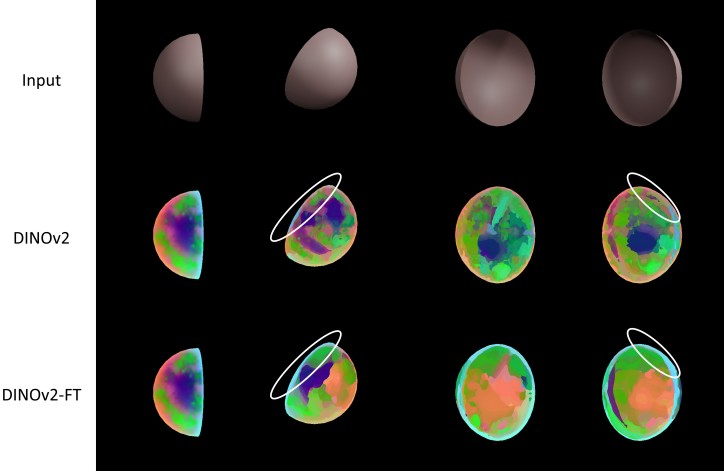

Figure 4: **Feature visualization of an untextured hemisphere from different viewpoints. Top row:** Input hemisphere rendered from four different angles. **Middle row:** Feature embeddings from DINOv2 visualized using RGB mapping, showing inconsistent features across views and edges (highlighted by white circles). **Bottom row:** Our fine-tuned DINOv2 produces more consistent features that better preserve correspondences across viewpoints, particularly at edges and inward outward views.

| Method | #Inliers |
|---|---|
| DINOv2 Oquab et al. (2023) | 99 |
| Finetuned | **159** |
| DINOv2-Reg Darcet et al. (2023) | 76 |
| Finetuned | **148** |
| MAE He et al. (2022a) | 97 |
| Finetuned | **196** |
| CLIP Radford et al. (2021) | 18 |
| Finetuned | **61** |
| DeiT Touvron et al. (2022) | 25 |
| Finetuned | **81** |

Table 13: **Quantitative results on the number of feature inliers that are background-invariant.**

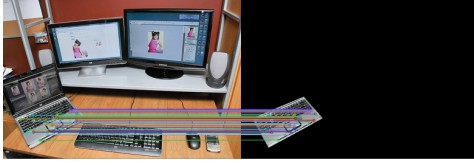

Before fine-tuning

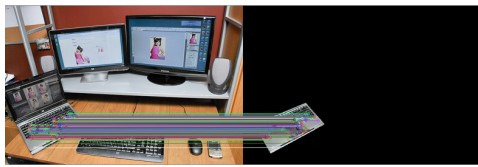

After fine-tuning

Figure 5: **Visualization of DINOv2's feature correspondence before and after fine-tuning, using mutual nearest neighbor.** After finetuning, we get more feature correspondences.

### A.11 QUALITATIVE RESULTS FOR POSE ESTIMATION/TRACKING/SEMANTIC TRANSFER

In this section, we provide qualitative comparisons on various downstream tasks. The results for pose estimation, tracking, and semantic correspondence are shown in Figures 10, 11 and 12, respectively. Since DINOv2-Reg exhibits performance highly similar to DINOv2, we omit its qualitative results.

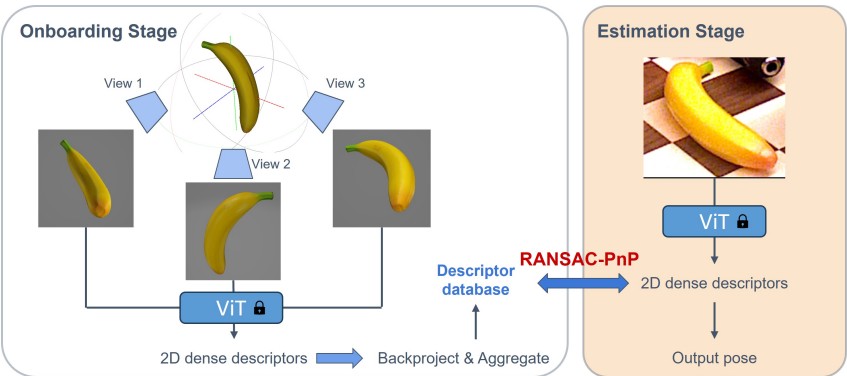

Figure 6: **Pose estimation pipeline.** During the onboarding phase, 2D dense features are extracted from the provided reference video and stored in a database. During inference, features are matched between a single query image and the database, followed by 3D-2D RANSAC-PnP to compute the final pose.

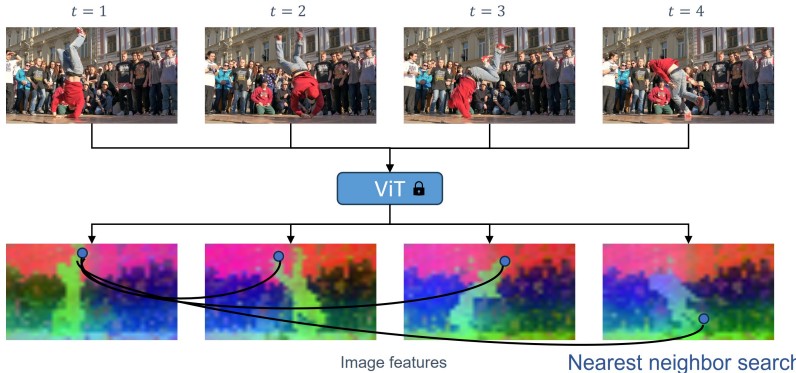

Figure 7: **Tracking pipeline.** For each point in the source frame, its nearest neighbors are located in the feature space across other frames.

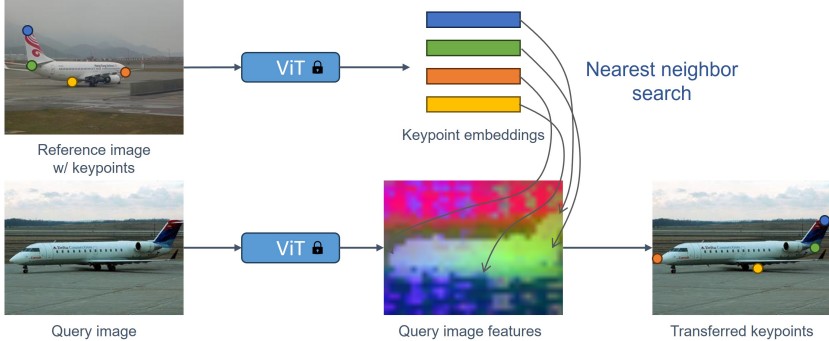

Figure 8: **Semantic transfer pipeline.** For the given keypoints in the reference image, descriptors are extracted using the frozen ViT, and their nearest neighbors are identified in the query image's feature space.

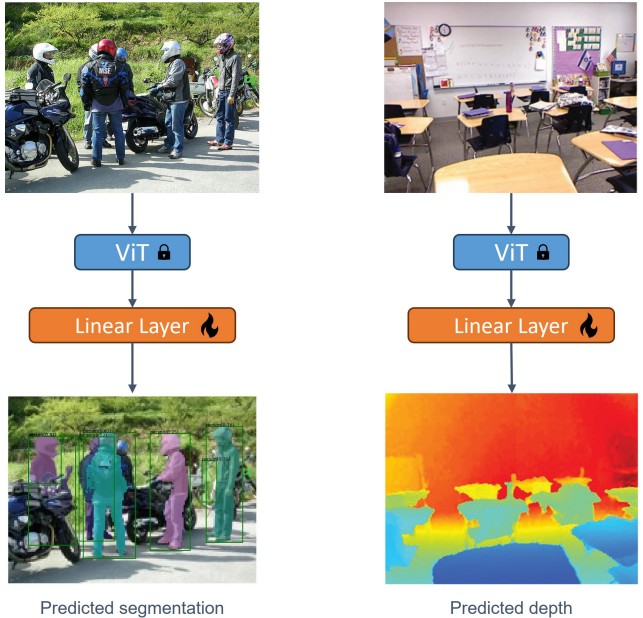

Figure 9: **Semantic segmentation and depth estimation pipeline.** Given an input image, a linear layer is fine-tuned on top of the frozen ViT to predict segmentation or depth.

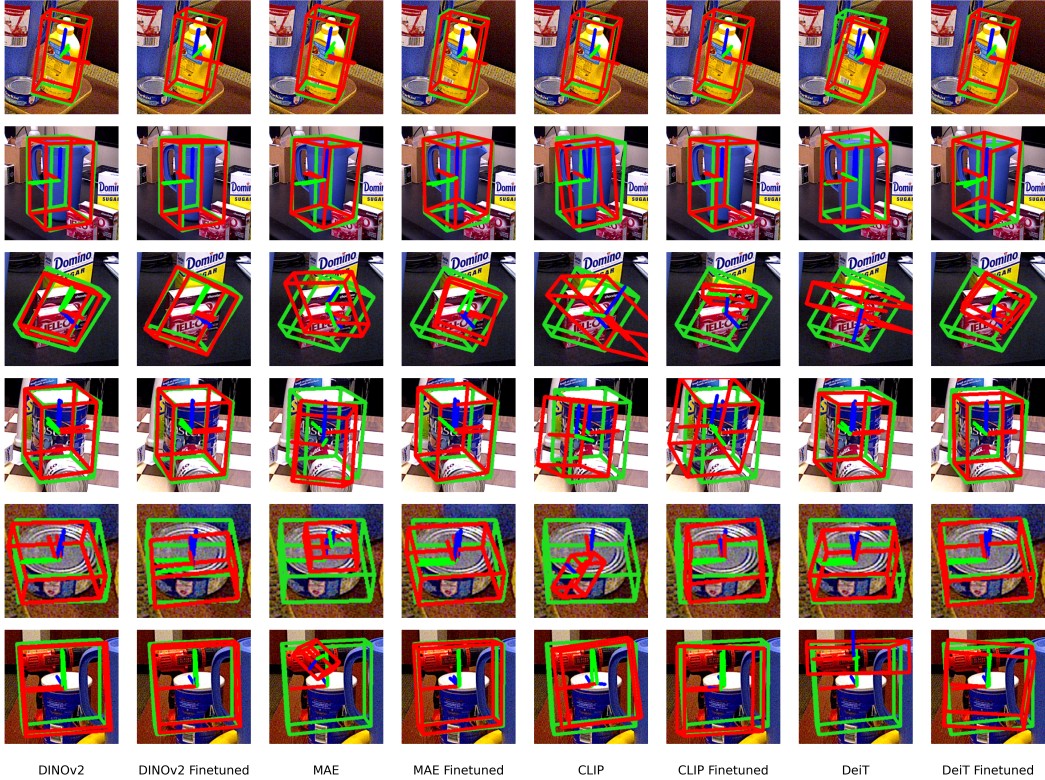

Figure 10: Qualitative results on YCB-Video pose estimation for different models, both before and after finetuning, are presented. Ground-truth poses are shown in green, while predictions are depicted in red. It can be observed that, in most cases, pose accuracy improves after finetuning, particularly for the MAE, CLIP, and DeiT models.

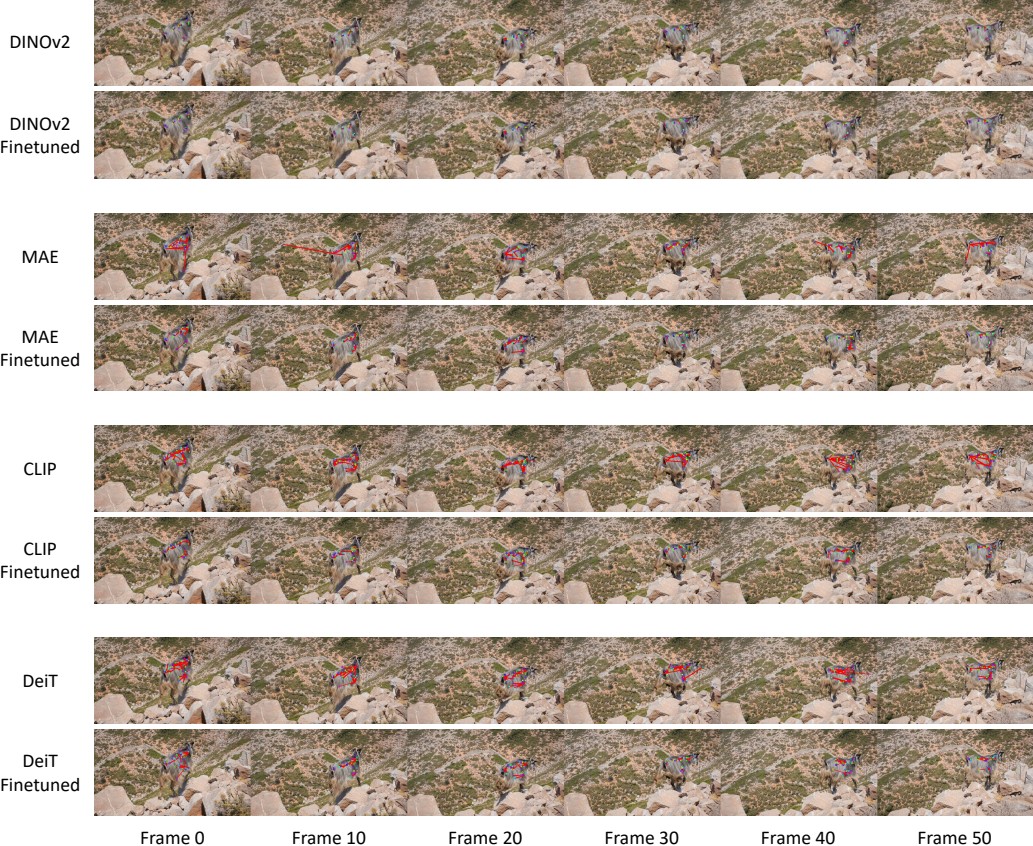

Figure 11: Qualitative results on TAP-VID-DAVIS for different models, both before and after fine-tuning, are shown. Query points are marked in various colors in the first frame, with red lines indicating the trajectory of the points. Prior to finetuning, the trajectories are highly noisy and inconsistent. However, after finetuning, tracking becomes significantly more stable.

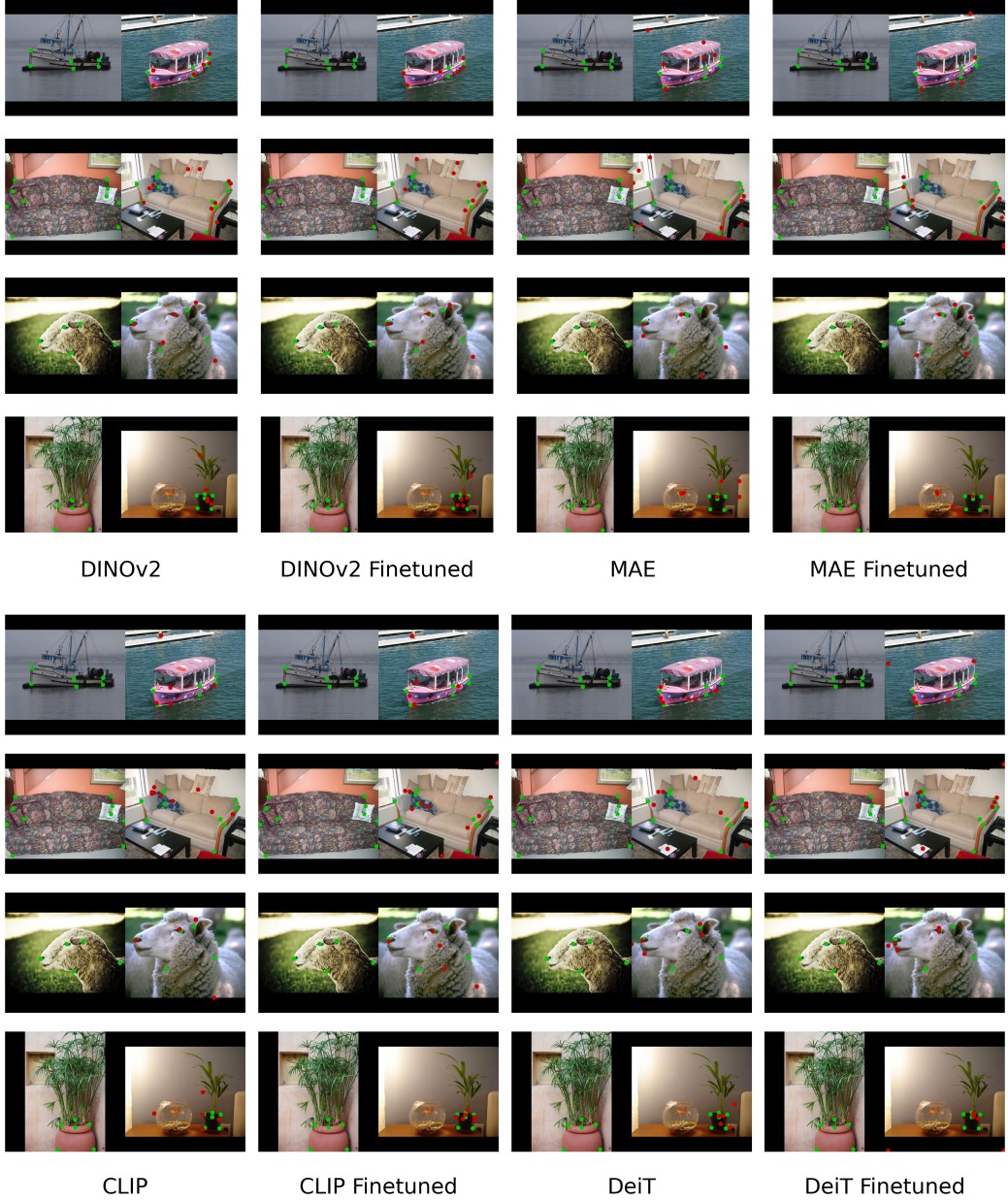

Figure 12: Qualitative results on PF-PASCAL (different views) for various models, both before and after finetuning, are presented. For each pair, the left image is the reference, and the right is the query. Ground-truth correspondences are shown in green, while predictions are depicted in red. It can be observed that, in most cases, finetuning improves accuracy by aligning the keypoints closer to their correct positions.