# OpenReview forum: "Multiview Equivariance Improves 3D Correspondence Understanding with Minimal Feature Finetuning"
_ICLR.cc/2025/Conference — ICLR 2025 Poster_

### Official Review · Reviewer_unL9 · 2024-10-23

**Soundness:** 3
**Presentation:** 3
**Contribution:** 3
**Rating:** 6
**Confidence:** 3

**Summary:**

This paper answers 3 questions: 1) To what extent do these models possess an inherent awareness of 3D structures? 2) How
does this awareness impact their performance on image-based 3D vision tasks? 3) Can we further
enhance the 3D awareness of these vision foundation models?
To answer the first question, the authors evaluate the pixel error of multi-view correspondence by using 5 pretrained vision models, and show DINOv2 has the strongest multiview equivalence. For the second question, the authors evaluate the performance of these models on 3 downstream tasks (pose estimation, video tracking, and semantic correspondence). Finally, the author proposes a simple finetuning strategy by enforcing feature similarity between corresponding views of a rendered synthetic object. Consistent improvements on downstream tasks are shown compared to the model without finetuning.

**Strengths:**

1. The paper is clearly organized and easy to follow. The figures and tables are also very clear and easy to understand.
2. The experiments are pretty comprehensive, covering 5 popular pretrained vision models, and 3 downstream tasks. Some interesting settings such as fine-tuning with only one object and with only 1 iteration are covered.

**Weaknesses:**

1. My main concern regarding this paper is about the limited practical use case of the finetuned DINO features. On the one hand, the fine-tuned feature alone is not very useful, as its performance on downstream tasks is much worse than the SOTA models (see Tab4 and Tab5 in the supplement, e.g., for point tracking, AJ=46.85 compared to Co-Tracker=65.6). On the other hand, there is no evidence showing that the fine-tuned feature this way will benefit downstream tasks if task-specific training/finetuning is performed. For example, the paper will be much more convincing to me if the fine-tuned DINO, as pre-trained weight, could be used to achieve SOTA performance on any specific vision task.
2. The conclusion that the authors draw to the first question, i.e., "To what extent do these models possess an inherent awareness of 3D structures", is not convincing. Having feature equivariance doesn't imply 3D awareness: SIFT could also match the same keypoint across views, even more accurately, but can you say SIFT has 3D awareness? The correspondence could be just coming from 2D local patch statistics, and no 3D reasoning is needed.
3. The evaluation protocol for the downstream tasks is not explained very clearly. For example, L160, you mentioned "during training" and "during inference", what is the training target and how long has the model been trained for? L214 in supp. How do you do monocular depth estimation with pretrained features? Do you need any additional training (e.g., using linear probing)? I think it would be beneficial if the authors could demonstrate the input/output of each task with figures.
4. There is no comparison with baselines on 3D-aware fine-tuning in the main body of the paper. However, there are comparisons with FiT in the supp, which I think is important and should be moved to the main paper. I have questions regarding the details of this comparison, see the next section.

**Questions:**

1. The authors compared the FiT baseline in Tab 3~7 in the supplement. I have some trouble understanding the result. 1) What is the difference between FiT and FiT-Reg? 2) The FiT results are significantly worse than the DINOv2 baseline. Since FiT is also based on DINO, this result is counterintuitive. The authors are basically claiming that the FiT fine-tuning is very harmful for all the tasks. Could the authors provide more evidence (e.g., visualizations) and discussion on this? I think FiT is an important baseline so this comparison will greatly affect my judgement of this paper.
2. The 3D correspondence is sometimes ambiguous, for example when the object is symmetric, or when self-occlusion occurs and the corresponding point is occluded. Have the authors deal with these cases explicitly? Would this ambiguity harm your model's performance?

---

> ### Author Response · Authors · 2024-11-24
>
> ## Q1: Limited practical use case of the finetuned DINO features?
> We would like to clarify that on OnePose-LowTex, our DINO fine-tuned method already outperforms OnePose++ on 3cm 3deg and 5cm 5deg metrics. However, on other datasets and tasks, the fine-tuned ViT still has some gap with state-of-the-art methods.
>
> On one hand, we expect better ViT architectures to emerge, which will yield improved performance after our fine-tuning (as our method is agnostic to the particular ViT architecture).  On the other hand, we focus on general-purpose ViT features that are more applicable than those domain specific features, and this paper’s main goal is not to beat the baselines.
>
> The key advantage of these ViT features is their **generality across different datasets and tasks**. They can be applied to a wide range of scenarios. For example: SparseDFF[1] uses DINO to aggregate and fine-tune consistent feature representations across views for few-shot transfer manipulation policy learning; LERF[2] uses dense DINO features for regularization; Wild Gaussians[3] employs off-the-shelf DINO features as a strong prior to estimate occlusions and reconstruct 3D scenes in the wild. These tasks lack clear methods for training domain-specific structures, as they are open-set tasks with limited training data or demonstrations. Therefore, we believe studying these general-purpose ViT features remains promising.
>
> To show our finetuned features can be useful in these general tasks, we conducted experiments on Wild-Gaussians and found that replacing the original features with our fine-tuned DINO features improved novel view synthesis quality in the wild, as shown in the following table. All results were produced using Wild-Gaussians' official GitHub repository.
>
> |                |  |Mountain       |         | |Fountain                      |         | | Corner                       |         | |Patio                         |         | |Spot                          |         |  | Patio-High                  |         |
> |----------------|----------------------|---------|---------|----------------------|---------|---------|----------------------|---------|---------|----------------------|---------|---------|----------------------|---------|---------|----------------------|---------|---------|
> |                | PSNR↑  | SSIM↑| LPIPS↓| PSNR↑              | SSIM↑  | LPIPS↓ | PSNR↑              | SSIM↑  |LPIPS↓ | PSNR↑              | SSIM↑  | LPIPS↓ | PSNR↑              | SSIM↑  | LPIPS↓ | PSNR↑              | SSIM↑  | LPIPS↓ |
> | Wild-Gaussian  | 20.82               | 0.668   | 0.239   | 20.90               | 0.668   | 0.213   | 23.51               | 0.810 | 0.152   | **21.31**               | 0.802   | 0.134   | 23.96               | 0.777   | 0.165   | 22.04               | 0.734   | 0.202   |
> |Ours           | **21.01**           | **0.672** | **0.234** | **20.97**           | **0.672** | **0.212** | **23.74**           | 0.810   | **0.151** | 21.23           | 0.802 | **0.133** | **24.01**           | **0.778** | **0.163** | **22.11**           | 0.734 | **0.201** |
>
>
> Additionally, we visualized LERF 3D features after replacing its DINO regularizer with our fine-tuned version. When given the text query "plate", LERF with our fine-tuned DINO produced a more focused and accurate relevancy map compared to the original DINO features, with better localization of the plate region and reduced noise in irrelevant areas such as cookies, as shown [in this image](https://ibb.co/3rhtKMB). We could only provide qualitative results for LERF since LERF has not released its quantitative evaluation code.
>
> We also included these discussion in our supplementary (Section A.7).
>
> ## Q2: Clarify why feature equivariance implies 3D awareness.
> We agree that we should clarify SIFT as 2D affine invariant rather than 3D-aware. SIFT, by design, is robust only to 2D transformations (rotation, translation, scale). It can only match keypoints across views with small changes, as the 2D image patch distorts minimally in such cases. To illustrate this, we have demonstrated [in this figure](https://ibb.co/PF4TMsB) that under large viewpoint changes in 3D, our fine-tuned DINO (DINOv2-FT) features give much better correspondences on MVImgNet, while SIFT fails due to significant image patch distortion. This comparison indicates that ViTs possess better 3D awareness than SIFT descriptors and exhibit some 3D understanding ability. As our evaluation in based solely on 3D feature equivariance, it does imply 3D awareness.
>
> [1] Sparsedff: Sparse-view feature distillation for one-shot dexterous manipulation.
>
> [2] Lerf: Language embedded radiance fields.
>
> [3] Wildgaussians: 3d gaussian splatting in the wild.

---

> ### Author Response · Authors · 2024-11-24
>
> ## Q3: Clarify evaluation protocal.
> Sorry for the confusion. None of the three downstream tasks in our main paper require training—they all directly use the **same** fine-tuned 3D-aware features. What we refer to as "training" on L160 is actually the onboarding phase, where we extract 2D dense features from the provided reference video and store them in our database. During inference, we match features between a single query image and our database.
>
> For depth estimation, we follow DINOv2's protocol by adding a linear layer with classification loss to predict depths across 256 uniform bins. For instance recognition and semantic segmentation, we also adhere to DINOv2's evaluation protocol.
>
> Thanks for the valuable suggestion, we will draw figures to illustrate the input and output for each task.
>
> ## Q4: Move comparisons with FiT into the main paper.
> Thanks for your insight. As our main goal is to show the fine-tuning of proximal rigid multi-view object centric tasks can improve downstream 3D understanding tasks, we did not put it into the main paper due to the space limit. But we added some text describing their performance in our revised manuscript (Section 3.2).
>
> ## Q5: Why FiT is so bad? Clarifications on FiT.
>
> FiT-Reg refers to FiT with DINOv2's registers[4]. Our experiments revealed that although FiT aims for 3D consistency, it significantly disrupts the semantics of certain parts, as shown [in this figure](https://ibb.co/YhCsLQb) and [this figure](https://ibb.co/qR8q46d). While this semantic disruption may be acceptable for FiT's original tasks like semantic segmentation and depth estimation—where an additional linear head can correct these issues—it becomes problematic for our tasks that require 3D-consistent, dense, pixel-level features. We hypothesize that FiT's poor performance stems from its naive approach to learning 3D consistency through an explicit 3D Gaussian field. When outliers or noise are present, the simple mean square error causes feature representations to shift toward these outliers.
>
> ## Q6: Occlusion and symmetric objects' influence on the performance.
>
> During fine-tuning, we handle self-occlusion by performing depth tests and discarding occluded samples.
>
> For symmetric objects, we don't implement specific handling—instead, we rely on uniform point sampling, where symmetric features' gradients cancel each other out in our SmoothAP loss, leaving the loss dominated by features from distinct parts. From our Figure 1 in the main manuscript, we can see that symmetric parts share similar embeddings. While there is one minimum requirement: we cannot have all points on an object symmetric (like a perfect sphere), otherwise the model cannot learn any meaningful features.
>
> [4] Vision Transformers Need Registers

---

> > ### Author Response · Authors · 2024-11-24
> >
> > Dear reviewer, we have included the illustration figure for each task in our updated supplementary (A.10).

---

> ### Comment · Reviewer_unL9 · 2024-11-25
> **Respond to the rebuttal**
>
> I appreciate the efforts the authors put into the rebuttal. The additional experiments and clarifications (e.g., the diagrams explaining how each task is evaluated) make the paper clearer and more convincing. The wild-Gaussian experiment is very interesting and should be presented in the main paper. It is also interesting to see that SIFT fails in extreme viewpoint while the proposed method is comparbly robust. A remaining concern (as agreed by other reviewers) is the inferior performance compared to SOTA on some tasks.
> Considering all these factors, I decide to increase my score from 5 to 6.
> A side note: the current version of the paper is 11 pages, which violates the authors guideline (https://iclr.cc/Conferences/2025/AuthorGuide). Please shrink it to 10 pages to avoid potential issues.

---

> > ### Author Response · Authors · 2024-11-26
> >
> > Thank you for your valuable feedback. In the updated manuscript, we have incorporated the Wild-Gaussian experiment into Section 3.4 as suggested. Additionally, we have shrinked the paper to 10 pages to align with the submission requirements.

---

### Official Review · Reviewer_A2uw · 2024-11-01

**Soundness:** 3
**Presentation:** 4
**Contribution:** 3
**Rating:** 6
**Confidence:** 4

**Summary:**

This paper evaluates the 3D awareness of ViT-based models and later proposes a strategy to improve 3D equivariance with minimum feature finetuning. The tasks for evaluating 3D awareness of ViT-based models are one-shot object pose estimation, video tracking, and semantic correspondence, and features from DINOv2, DINOv2-Reg, MAE, CLIP and DeiT are evaluated. Experimental results show that with some simple strategy of finetuning the foundation models, their 3D awareness can be improved with an obvious margin and the foundation model features can have better multi-view equivariance.

**Strengths:**

++ This paper is very well motivated and super clearly written. The paper first starts from evaluating the capability of vision foundation models on understanding 3D structures. Then, the paper shows the strong correlation between the multi-view equivariance and the performance of the chosen downstream tasks, demonstrating the reason for improving multi-view equivariance. Finally, the paper proposes a solution to improve the multi-view equivariance and therefore on the downstream tasks. The workflow of this paper is very natural and easy to understand.

++ The proposed solution to improve multi-view equivariance is simple but effective. With simply learning the equivariance on two views from the objects in Objaverse, the multi-view equivariance can be improved, so as their performance on downstream tasks.

++ The experimental results are extensive and clearly presented mainly in the forms of figures (e.g., Figures 3-4, Figures 6-10), to clearly show the improvement from employing the proposed feature finetuning method.

++ It is a very interesting and inspirational finding in Section 3.3 that only tuning the model with a single multi-view pair of one object for a single iteration can significantly boost the multi-view equivariance of foundation models.

**Weaknesses:**

-- I think the biggest weakness is that there is no recent methods for comparison on the chosen downstream tasks for reference. I did not mean that the performance of foundation models need to beat the current state-of-the-art on these specific tasks, but it is necessary to provide these comparisons to give readers a sense of how good foundation models can achieve in performance. If performance from foundation models are far away behind the current state-of-the-arts, then there will be less need or motivation for future research to work on employing foundation models for 3D tasks.

-- This paper only studied the final-layer features from the vision transformer models. However, in other works that study the representation capability of features like LPIPS [1] or DVT [2], features from multiple layers are studied. Are there any reasons for studying the final-layer features? Otherwise this study would lose generalizability.

[1] Zhang et al. The Unreasonable Effectiveness of Deep Features as a Perceptual Metric. CVPR 2018.

[2] Yang et al. Denoising Vision Transformers. ECCV 2024.

**Questions:**

-- This paper mainly studies ViT-based foundation models. Are there any reasons not to study foundation models with other architectures, like ConvNeXt [1]? Is it because in current days ViT-based models are the most commonly used ones? And do the authors expect non-ViT-based models to have similar behaviour as ViT-based models?

-- In Lines 363-364, the paper mentions that "even simple shapes like an untextured hemisphere can enhance 3D understanding". However, I do not find this point reasonable (although this might be supported by experimental evidence). In principle, an untextured hemisphere would be rotation-invariant when the viewing angles rotate in certain directions, due to the symmetry of the shape, which makes the features at the points on the same radius to be consistent. How could the model actually learn 3D understanding from such a hemisphere shape? I am not sure whether this can be shown from some visualizations of the learned feature maps of a hemisphere shape.

[1] Liu et al. A ConvNet for the 2020s. CVPR 2022.

---

> ### Author Response · Authors · 2024-11-24
>
> ## Q1: Comparison with state-of-the-art task-specific tasks.
> Thank you for the suggestion. We indeed included baseline methods for pose estimation and tracking in our supplementary materials (Section A.3). For OnePose-LowTex, our DINO fine-tuned method already outperforms OnePose++ on 3cm 3deg and 5cm 5deg metrics. However, on other datasets and tasks, the fine-tuned ViT still has some gap with state-of-the-art methods.
>
> On one hand, we expect better ViT architectures to emerge, which will yield improved performance after our fine-tuning (as our method is agnostic to the particular ViT architecture).
>
> On the other hand, as you mentioned, these ViT features aren't designed to surpass domain-specific methods—a point also illustrated in DINOv2 paper’s table, which only compares general-purpose feature learning methods.
>
> The key advantage of these ViT features is their **generality across different datasets and tasks**. They can be applied to a wide range of scenarios. For example: SparseDFF[1] uses DINO to aggregate and fine-tune consistent feature representations across views for few-shot transfer manipulation policy learning; LERF[2] uses dense DINO features for regularization; Wild Gaussians[3] employs off-the-shelf DINO features as a strong prior to estimate occlusions and reconstruct 3D scenes in the wild. These tasks lack clear methods for training domain-specific structures, as they are open-set tasks with limited training data or demonstrations. Therefore, we believe studying these general-purpose ViT features remains promising.
>
> To show our finetuned features can be useful in these general tasks, we conducted experiments on Wild-Gaussians and found that replacing the original features with our fine-tuned DINO features improved novel view synthesis quality in the wild, as shown in the following table. All results were produced using Wild-Gaussians' official GitHub repository.
>
> |                |  |Mountain       |         | |Fountain                      |         | | Corner                       |         | |Patio                         |         | |Spot                          |         |  | Patio-High                  |         |
> |----------------|----------------------|---------|---------|----------------------|---------|---------|----------------------|---------|---------|----------------------|---------|---------|----------------------|---------|---------|----------------------|---------|---------|
> |                | PSNR↑  | SSIM↑| LPIPS↓| PSNR↑              | SSIM↑  | LPIPS↓ | PSNR↑              | SSIM↑  |LPIPS↓ | PSNR↑              | SSIM↑  | LPIPS↓ | PSNR↑              | SSIM↑  | LPIPS↓ | PSNR↑              | SSIM↑  | LPIPS↓ |
> | Wild-Gaussian  | 20.82               | 0.668   | 0.239   | 20.90               | 0.668   | 0.213   | 23.51               | 0.810 | 0.152   | **21.31**               | 0.802   | 0.134   | 23.96               | 0.777   | 0.165   | 22.04               | 0.734   | 0.202   |
> |Ours           | **21.01**           | **0.672** | **0.234** | **20.97**           | **0.672** | **0.212** | **23.74**           | 0.810   | **0.151** | 21.23           | 0.802 | **0.133** | **24.01**           | **0.778** | **0.163** | **22.11**           | 0.734 | **0.201** |
>
>
> Additionally, we visualized LERF 3D features after replacing its DINO regularizer with our fine-tuned version. When given the text query "plate", LERF with our fine-tuned DINO produced a more focused and accurate relevancy map compared to the original DINO features, with better localization of the plate region and reduced noise in irrelevant areas such as cookies, as shown [in this image](https://ibb.co/3rhtKMB). We could only provide qualitative results for LERF since LERF has not released its quantitative evaluation code.
>
> We also included these discussion in our supplementary (Section A.7).
>
> [1] Sparsedff: Sparse-view feature distillation for one-shot dexterous manipulation.
>
> [2] Lerf: Language embedded radiance fields.
>
> [3] Wildgaussians: 3d gaussian splatting in the wild.

---

> > ### Author Response · Authors · 2024-11-24
> >
> > ## Q2: Multi-layer feature fusion ablation.
> > Thank you for your valuable advice. Indeed, we can utilize multi-layer representations. Specifically, we experimented with two different variations: concatenating the features from the last 4 layers and concatenating features from the 2nd, 5th, 8th, and 11th layers. The results are presented in the Table. We found that fusing features from different layers does improve the instance-level correspondence a little bit but greatly harms semantic correspondences in tracking and semantic transfer. This indicates that features from earlier layers focus more on instance-level details, while the final layer captures more semantic information. We've included this analysis in our supplementary materials (A.8.1).
> >
> > |                        |      |  OnePose-LowTex            |         |      |      TAP-VID-DAVIS       |         |           |     PF-PASCAL      |         |
> > |------------------------|:-----------------------:|:-------:|:-------:|:---------------------:|:-------:|:-------:|:---------------------:|:-------:|:-------:|
> > |                        | 1cm 1deg               | 3cm 3deg| 5cm 5deg| AJ                   | δ_avg   | OA      | PCK0.05              | PCK0.10 | PCK0.15 |
> > | DINOv2-single-scale    | 13.58                  | 58.03   | 77.35   | **46.85**            | **63.84** | **84.15** | **47.24**            | **60.76** | **67.57** |
> > | DINOv2-2,5,8,11        | **15.34**              | 59.56   | 76.81   | 39.67               | 56.74   | 76.29   | 39.84               | 53.05   | 60.15   |
> > | DINOv2-8,9,10,11       | 14.24                  | **60.35** | **79.27** | 41.25               | 56.56   | 80.15   | 44.99               | 57.73   | 64.48   |

---

> ### Author Response · Authors · 2024-11-24
>
> ## Q3: Results on other models with different architectures like ConvNext.
> Thank you for your insightful comment. We have applied our method to other architectures like ConvNeXt and found that we can consistently improve its performance on downstream tasks as well. However, we've also observed that ConvNeXt features are not as good as those of modern ViTs. This is one of the main reasons we chose to focus on ViT-based models—they are not only the most commonly used but potentially superior. Overall, we do expect and observe improvements in non-ViT based methods like ConvNeXt. This finding is particularly interesting as it teaches us a valuable lesson: with relatively simple 3D fine-tuning, we can achieve even better 3D features than those obtained through pretraining on a vast set of unstructured 2D images. We included this experiment in our supplementary (A.5).
> | | | OnePose-LowTex |  |  | TAP-VID-DAVIS |  |  | PF-PASCAL |  |
> |-------------------|----------|----------|----------|-------|--------|-------|---------|---------|---------|
> |                   | 1cm 1deg | 3cm 3deg| 5cm 5deg | AJ    | δ_avg  | OA    | PCK0.05 | PCK0.10 | PCK0.15 |
> | ConvNext-small    | 3.25     | 13.46    | 21.39    | 15.98 | 26.08  | **74.72** | 10.32   | 16.30   | 22.17   |
> | small-finetuned   | **5.28** | **19.98**| **28.23**| **16.70** | **26.56** | 74.54  | **11.61** | **19.38** | **25.56** |
> | ConvNext-base     | 5.10     | 22.22    | 34.81    | 17.57 | 28.21  | **72.47** | 13.62   | 21.03   | 27.81   |
> | base-finetuned    | **8.05** | **32.69**| **46.41**| **18.53** | **28.48** | 71.24  | **15.64** | **25.37** | **32.13** |
> | ConvNext-large    | 4.71     | 25.33    | 36.48    | 19.43 | 30.24  | 73.71  | 11.05   | 17.57   | 24.19   |
> | large-finetuned   | **7.21** | **30.68**| **44.47**| **19.45** | **30.68** | **74.33** | **14.56** | **24.04** | **31.57** |
>
> ## Q4: How can the model learn from hemisphere shape?
>
> Unlike a perfect sphere, the hemisphere we used is not completely symmetric and provides information about edges and viewpoint orientation. [Our visualization of the learned embeddings](https://ibb.co/QdvPBmw) shows that after fine-tuning on the hemisphere, the network achieves better edge correspondences and can differentiate between inward and outward views. Even though the object lacks texture, the shadows and edge features provide sufficient cues for the ViT features to develop 3D understanding.
>
> Similarly, in cognitive science, scientists have discovered that the human brain also learns complex 3D structures from basic geometric primitives. Biederman's Recognition-by-Components (RBC) theory[4] suggests that humans recognize objects through simple 3D primitives called geons (geometrical ions)—basic shapes such as cubes, cylinders, and cones. We have included these discussions in our supplementary (A.9).
>
> [4] ecognition-by-components: a theory of human image understanding.

---

> ### Comment · Reviewer_A2uw · 2024-11-25
>
> Thanks the authors for providing the detailed rebuttal! I think most of my concerns are properly addressed (extracting features from other layers, other foundation models like ConvNeXt, the feature map for hemisphere shape). However, for my most major concern of comparing with recent methods, although the authors mention that in the pose estimation task on OnePose-LowTex, the proposed method surpasses the state-of-the-arts, in many other cases (like still the pose estimation task on YCB-Video, and what Reviewer unL9 has mentioned on the tracking task), the gap between the proposed method and the state-of-the-art is still very noticeable from my point of view.
>
> I do think the presentation and motivation of this paper is very excellent, as I find myself enjoyable reading through the whole paper, but the sub-optimal performance compared to state-of-the-arts (the performance gap is a bit large I think) prevents me from further raising my scores, so I think 6 is a reasonable score for now.

---

### Official Review · Reviewer_XtFz · 2024-11-02

**Soundness:** 2
**Presentation:** 3
**Contribution:** 2
**Rating:** 6
**Confidence:** 4

**Summary:**

In this work the authors studied the importance of multiview equivariance for the tasks of pose estimation, video tracking, and semantic correspondence. Results show that vision models with better multiview equivariance also achieve better performance for the three downstream tasks. Moreover, by finetuning the model on synthetic multi-view images, models with better equivariance perform better on various tasks.

**Strengths:**

1. The authors studied the multiview equivariance property of vision foundation models, and associate it with the performance of three downstream tasks. This enable a more systematic way to analyze the part correspondence of vision models and help to understand the limitations of models on downstream tasks.
2. The authors proposed to finetune the model with multi-view synthetic images, improving multiview equivariance and downstream tasks. This proposed approach is straightforward but demonstrated effective on downstream tasks.

**Weaknesses:**

1. The three tasks considered in this paper, (keypoint-based) pose estimation, video tracking, and semantic correspondence, are all ultimately part correspondence problem, which benefits from multiview equivariance. The title and introduction gives the impression that multiview equivariance improves 3D understanding in general, but truly the experiments only focused on very specific tasks.
2. I understand that pose estimation is a 3D understanding problem, but I don't think video tracking and semantic correspondence falls into the picture of 3D understanding, given the title of the paper. Specifically how this paper fits into the analysis of 3D awareness considering previous works [A,B].
3. The authors argued the importance of multiview equivariance on tasks such as pose estimation, video tracking, and semantic correspondence. This is only partially true as it also depends on the nature of the algorithm, bottom-up or top-down. For instance, [B] studied the 3D awareness of vision foundation models for pose estimation. Vision-language models often learns a top-down representation for 2D/3D recognition so view equivariance could hurt the performance in such cases. The authors should address these points to reinforce the integrity of the paper.

[A] Probing the 3d awareness of visual foundation models.
[B] ImageNet3D: Towards General-Purpose Object-Level 3D Understanding.

**Questions:**

The authors could provide some clarifications on **[W2]** and **[W3]**.

---

> ### Author Response · Authors · 2024-11-24
>
> ## Q1: General 3D understanding and correspondence tasks relationship.
> Thank you for this insightful observation. Correspondence estimation is a fundamental component of 3D vision understanding, underlying key tasks such as epipolar geometry, stereo vision for 3D reconstruction, and optical flow or tracking to describe the motion of a perceived 3D world. Stereo cameras, and even human perception, rely on disparity maps—effectively, correspondences between projected 3D parts to understand depth and spatial relationships.
>
> The three tasks we evaluated—pose estimation, video tracking, and semantic correspondence—were intentionally selected to cover diverse aspects of correspondence estimation, ranging from simpler to more complex scenarios:
>
> 1. Pose Estimation examines correspondences within the same instance under rigid transformations (SE(3));
>
> 2. Video Tracking extends this to correspondences for the same instance under potential non-rigid or articulated transformations, such as humans or animals in motion;
>
> 3. Semantic Correspondence requires correspondences across different instances with similar semantics, often under arbitrary transformations.
>
> An qualitative illustration of these three different correspondences is shown [in this link](https://ibb.co/yqmmLmL). We've also included more discussion in Section 2.1.2 to clarify these distinction.
>
> A key contribution of our work is demonstrating that finetuning models using a simple SE(3) correspondence setup during training enables them to generalize across all three tasks, i.e., correspondence types at test time. This result highlights the non-trivial ability of vision models to extrapolate learned multiview equivariance to more complex and diverse scenarios.
>
> To address the reviewer's concern, we are open to revising the title to more explicitly reflect our focus on 3D correspondences, ensuring it better aligns with the scope of our experiments and contributions. Additionally, we have expanded the discussion in Section 2.1 to provide a deeper analysis of the relationship between these tasks and 3D correspondence. Please let us know if an updated title is needed.
>
> ## Q2: Inclusion of video tracking and semantic correspondence, and how this work relates to prior works A and B.
> Thank you for your observation. We recognize the need to clarify how the selected tasks—pose estimation, video tracking, and semantic correspondence—fit into the broader scope of 3D understanding and how our work relates to previous studies. As discussed in our response to the previous question, these tasks were chosen to evaluate different aspects of correspondence estimation, a critical capability in 3D vision.
>
> Regarding prior works:
> - **[A]** explored multiview geometry correspondences, similar to our evaluation in Section 2. However, their experiments were conducted on relatively small datasets, such as NAVI (36 objects) and ScanNet paired views (1500 test pairs). In contrast, we used a large-scale dataset with approximately 1M image pairs, enabling more robust conclusions on multiview correspondence. More importantly, while [A] focused on evaluation, we went one step further and proposed a simple and effective fine-tuning approach that enhances 3D correspondence capabilities in vision models, demonstrating its generalization across multiple downstream applications.
> - **[B]** studied a related but distinct problem—whether global tokens (up-down representations) vary across different views. Their work explores a complementary area to ours, as they focus on view-dependent global features, whereas we emphasize dense, pixel-level features that are invariant to viewpoint changes. Our results highlight the utility of these dense features for a variety of applications.
> See the next question for more discussion about B.
>
> We have expanded the discussion in the introduction and related work sections to better articulate the positioning of our work within the 3D awareness landscape and how it relates with prior studies.

---

> > ### Author Response · Authors · 2024-11-24
> >
> > ## Q3: Multiview equivariance can hurt tasks in top-down approaches; how equivariance can help in general.
> > Thank you for raising this important point. We agree that the impact of multiview equivariance depends on the algorithm's approach, whether it is bottom-up or top-down, and appreciate the opportunity to discuss this distinction further.
> >
> > [B]'s top-down approach to pose estimation relies on classifying different poses using pretrained features with an added domain-specific linear layer. However, as our paper focuses on general-purpose ViT features, it is hard to apply [B]'s method across different dataset domains like OnePose-LowTex and YCB-Video. Consider a novel object: how would [B]'s method determine whether image A represents pose A and image B represents pose B, when these pose label meanings were predefined during training? Without a well-defined canonical pose for the novel object, image A could equally represent pose B.
> >
> > For general unseen tasks and datasets, we argue finding correspondences—or equivalently, learning equivariant representations, is a better approach. Features that vary across viewpoints are unsuitable for general-purpose settings since we neither know nor can control their variation. Similarly, it is unclear how to do video tracking and semantic transfer, using varying features, without training domain-specific heads. To reinforce the integrity of our paper, we have added this discussion to the related work.
> >
> > [A] Probing the 3d awareness of visual foundation models.
> >
> > [B] ImageNet3D: Towards General-Purpose Object-Level 3D Understanding.

---

> ### Comment · Reviewer_XtFz · 2024-11-25
> **Official Comment by XtFz**
>
> Thank the authors for the response. I think the revision has improved the clarity of the paper.
>
> One of my concerns aligns with the W2 of unL9. I agree that having 3D understanding would benefit tasks such as video tracking, e.g., in the case demonstrated in Figure 4, but in general I think the major challenges of video tracking are still truncation, occlusion, and changes in appearances, etc.
>
> I also agree with the W1 of unL9. I think the results achieved by finetuning on synthetic 3D models are interesting, but more analyses are needed, given how widely pretrained DINO has been adopted in various tasks and settings while it remains unclear how the proposed finetuning would affect downstream tasks.
>
> In general I acknowledge the contributions of this work. The presentation is improved with the revision and discussions about related works. I think some minor changes in the title would also help clarify the main focus of the paper.

---

> > ### Author Response · Authors · 2024-11-25
> >
> > Thank you for your swift reply.
> > ## Q1: Concerns about how 3D understanding boosts video tracking
> >
> > We acknowledge that video tracking is a multifaceted challenge involving various critical factors, while the primary focus of our paper is on 3D correspondence. Addressing other aspects, such as truncation and occlusion, is beyond the scope of this work. Our additional examples (shown [in this link](https://ibb.co/QMjyWPV)) from TAP-VID-DAVIS demonstrate that in video tracking, most appearance changes stem from viewpoint changes and deformations—complex procedures occurring in the 3D world. Traditional 2D understanding methods like SIFT descriptors fail completely in these scenarios. This is precisely why we chose video tracking as one of our downstream tasks.
> >
> > ## Q2: How the proposed finetuning affects downstream tasks
> >
> > Thank you for your comment and acknowledgment of the importance of studying DINO. We have included more analysis and discussions on how our finetuned DINO features can be applied to downstream tasks such as LERF and Wild-Gaussians. Our evidence demonstrates that our finetuned DINO benefits multiple downstream tasks, including the three tasks in our original paper, along with language-embedded neural fields (LERF) and occlusion-robust 3D reconstruction in the wild (Wild-Gaussians). For more details, please refer to our response to Q2 in the reply to unL9.
> >
> > ## Q3: Title change
> >
> > We appreciate your valuable suggestion regarding the title. We have changed it from "**Multiview Equivariance Improves 3D Understanding with Minimal Feature Finetuning**" to "**Multiview Equivariance Improves 3D Correspondence Understanding with Minimal Feature Finetuning**" in the updated manuscript, to better reflect the paper's focus on 3D correspondence.
> >
> >  If you have further questions or suggestions of experiments, please don't hesitate to let us know. We strive to address any outstanding questions.

---

> > > ### Comment · Reviewer_XtFz · 2024-11-25
> > > **Official Comment by XtFz**
> > >
> > > Thank the authors for the further clarifications. I will increase my rating to borderline accept.

---

### Official Review · Reviewer_f5uK · 2024-11-04

**Soundness:** 3
**Presentation:** 3
**Contribution:** 3
**Rating:** 6
**Confidence:** 4

**Summary:**

This paper evaluates 2D ViT-based foundation models' abilities to learn 3D equivariant features, shows the significance of 3D equivariant on 3D downstream tasks (pose estimation, video tracking, and semantic correspondence), and proposes a very simple finetuning strategy that boosts the 3D understanding abilities of these existed 2D foundation models by introducing 3D information from either synthetic data (Objaverse) or real data (MVImgNet).

**Strengths:**

1. The proposed finetuning strategy is simple to adopt and easily reproducible. Finetuning shows performance boost with even only one additional sample.

2. The experiments are comprehensive, covering three main downstream tasks of 3D equivariant features (one-shot object pose estimation, video tracking and semantic correspondence), highlighting the significance of 3D equivariant features.

3. The paper discusses finetuning using different types of data (synthetic data, real data and scenes), and conducts good ablations of the model's design (added conv head).

**Weaknesses:**

Minor concerns:
1. Table 1 ablates on number of added conv layers to a given ViT, and one additional conv head gives best performance boost instead of two or three. Some analysis of why this is happening will be nice.

2. The whole paper lacks some mathematical formulation and explanation. For example, there is no formula for two evaluation metrics define in the paper: APE and PCDP. Also, it does not have math for the loss (SmoothAP). Some additional math can be more followable than text.

**Questions:**

Please refer to the weakness part for my questions.

Overall, this works presents an interesting yet very simple method that is easily reproducible to make 2D vision model generates better 3D equivariant features, and I believe it can benefit the community.

---

> ### Author Response · Authors · 2024-11-24
>
> ## Q1: Why one conv layer performs better than multiple layers.
>
> Thank you for the suggestion. Upon analyzing the effect of additional convolutional layers, we found that while one additional convolutional layer significantly improves the performance, adding two or three layers introduces noise into the features. This noise likely arises from the increased parameter freedom, which can overfit to local patterns and reduce the consistency of dense pixel-wise features. To illustrate this, we included feature visualizations [in this link](https://ibb.co/rZH1S8d). These visualizations clearly show that the additional layers produce less coherent features, leading to a degradation in downstream task performance. This analysis has been added to the revised supplementary (A.8.3) for clarity.
>
> ## Q2: Mathematical formulas and explanations for APE, PCDP, and SmoothAP metrics.
>
> Thank you for pointing this out. We agree that including mathematical formulations for key metrics and the loss function would enhance the clarity and followability of our paper. To address this, we have added the following formulations for APE, PCDP, and SmoothAP to the revised supplementary (A.1).
>
> - **Average Pixel Error (APE)**: Suppose we have $N$ objects, each rendered from $k=42$ different views. For a pixel $x_1$ in the first image, the ground-truth corresponding pixel $x_2$ in the second image is determined via back-projection into 3D and re-rendering, excluding occluded points. The evaluated method predicts $\tilde{x}_2$. APE is computed as:
>
> $$
>         APE = \sum_N\sum_i^k\sum_j^k\sum_{x_1\rightarrow x_2}\frac{\|x_2 - \tilde{x}_2\|_2}{\min(W,H)}
>         $$
>
> where $W,H$ are the image width and height.
>
> - **Percentage of Correct Dense Points (PCDP)**: PCDP measures the proportion of predicted points $\tilde{x}_2$ that fall within a normalized threshold $\delta$ of the ground-truth point $x_2$:
>
> $$
> PCDP=\sum_N\sum_i^k\sum_j^k\sum_{x_1\rightarrow x_2}\mathcal{1}(\frac{\|x_2 - \tilde{x}_2\|_2}{\min(W,H)} < \delta)
> $$
>
> Here $\mathcal{1}(\cdot)$ is the indicator function and $\delta$ is a threshold (commonly 0.05, 0.1 or 0.15).
>
> - **Smooth Average Precision (SmoothAP)**: SmoothAP is used as the training loss to enforce accurate feature correspondences:
>
> $$
>             SmoothAP=\frac{1}{S_P}\sum_{i\in S_P}\frac{1+\sum_{j\in S_P}\sigma(D_{ij})}{1+\sum_{j\in S_P}\sigma(D_{ij})+\sum_{j\in S_N}\sigma(D_{ij})}
> $$
>
> where given a query point $x_1$, $S_P$ is the positive set containing ground-truth points $\{x_2\}$,$S_N$ is the negative set containing all other points in the second view, and $\sigma$ is the sigmoid function, and $D_{ij}=f_j\cdot f_{x_1} - f_i\cdot f_{x_1}$ measures the difference in feature similarity with respect to the query point $x_1$. Ideally, we want all negative points to have smaller similarities with respect to $x_1$ than all positive ones. In this case, $\sum_{j\in S_N}\sigma(D_{ij})=0$ and we get $SmoothAP=1$. In training, we optimize the loss: $1 - SmoothAP$.

---

### Author Response · Authors · 2024-11-24
**General Response**

We thank reviewers for their valuable feedback and recognition of our work. Specifically, we appreciate the acknowledgment of our paper **well motivated** [A2uw], **clearly written** [A2uw, unL9], **comprehensive experiments** [unL9, A2uw, f5u], and **innovative approach to improving multiview equivariance in vision foundation models** [XtFz, A2uw]. Below, we provide a detailed summary of key responses and updates.

---

### **[f5uK] Why one conv layer performs better than multiple layers**
We found adding more layers led to noise and overfitting, reducing feature consistency. To clarify this, we have included feature visualizations in the revised paper, showing that additional layers introduce artifacts that harm downstream performance.

---

### **[XtFz] General 3D understanding and correspondence tasks relationship**
Correspondence estimation is a fundamental component of 3D vision understanding, underlying key tasks such as epipolar geometry, stereo vision for 3D reconstruction, and optical flow or tracking to describe the motion of a perceived 3D world. Our experiments encompass diverse scenarios:
- **Pose Estimation:** Correspondences within an instance under rigid transformations (SE(3)).
- **Video Tracking:** Correspondences under non-rigid transformations, e.g., human motion, with arbitrary viewpoint changes.
- **Semantic Correspondence:** Correspondences across instances with similar semantics under arbitrary viewpoint changes.
---

### **[A2uw,unL9] Comparisons to SOTA; practical use of the finetuned DINO features**
- On **OnePose-LowTex**, our fine-tuned DINO outperforms OnePose++ on 3cm 3deg and 5cm 5deg metrics.
- On other datasets, while our results are not as good as domain-specific methods, our fine-tuned features improve 3D scene understanding in general-purpose applications, as evidenced by:
    - Improved performance in **novel view synthesis** (Wild-Gaussians).
    - Enhanced 3D alignment for **semantic queries** in LERF.

---

### **[A2uw,unL9] How can model learn from (partially-)symmetric hemisphere**
For symmetric objects, we don't implement specific handling—instead, we rely on uniform point sampling, where symmetric features' gradients cancel each other out in our SmoothAP loss, leaving the loss dominated by features from distinct parts. And symmetric parts will share similar embeddings as they are equally changable in the loss.

Besides, the hemisphere we used is not completely symmetric and provides information about edges and viewpoint orientation. Our [visualization of the learned embeddings](https://ibb.co/QdvPBmw) shows that after fine-tuning on the hemisphere, the network achieves better edge correspondences and can differentiate between inward and outward views.

### **[A2uw] Multi-layer feature fusion ablation**
We tested two different multi-layer representations. These improve instance-level correspondence slightly but degrade performance in semantic tasks. This tradeoff is now discussed in the supplementary materials.

---

### **[f5uK] Mathematical Formulations**
We have added mathematical definitions for the evaluation metrics (APE and PCDP) and loss function (SmoothAP) in the revised paper.

---


We hope these revisions address all concerns and improve the paper’s clarity and impact. Please refer to the detailed responses for more information.

---

### Meta-Review · Area_Chair_HTPc · 2024-12-18

**Metareview:**

The paper evaluates 3D equivariance in the features of foundation models, and shows the significance of 3D equivariance in 3D downstream tasks. Finally, it proposes a simple fine-tuning strategy that enhances the 3D equivariance in existing 2D foundation models using synthetic and real data, leading to improved results.

The paper is well-written the method is simple yet very effective. The experimental evaluation is comprehensive and demonstrates the effectiveness of the approach as well as the broader relevance of the research results.

One remaining concern of (Reviewers unL9 and A2uw) is the sub-optimal performance compared to state-of-the-art models on some tasks, but overall the experimental evaluation is convincing.

The reviewers and AC unanimously agree that the paper should be accepted.

**Additional Comments On Reviewer Discussion:**

The authors made a comprehensive rebuttal and all reviewers actively participated in the discussion. Overall, the rebuttal was succesful in convincing initially negative reviewers of the merit of this work.

---

### Decision · Program_Chairs · 2025-01-22

Accept (Poster)